# The force loading rate drives cell mechanosensing through both reinforcement and cytoskeletal softening

Ion Andreu[1,12], Bryan Falcones [2,12], Sebastian Hurst[3], Nimesh Chahare[1,4], Xarxa Quiroga [1,2], Anabel-Lise Le Roux [1], Zanetta Kechagia [1], Amy E. M. Beedle[1,5], Alberto Elosegui-Artola[1,6,7], Xavier Trepat[1,2,8,9], Ramon Farré[2,10,11], Timo Betz [3], Isaac Almendros [2,10,11✉] & Pere Roca-Cusachs [1,2✉]

Cell response to force regulates essential processes in health and disease. However, the fundamental mechanical variables that cells sense and respond to remain unclear. Here we show that the rate of force application (loading rate) drives mechanosensing, as predicted by a molecular clutch model. By applying dynamic force regimes to cells through substrate stretching, optical tweezers, and atomic force microscopy, we find that increasing loading rates trigger talin-dependent mechanosensing, leading to adhesion growth and reinforcement, and YAP nuclear localization. However, above a given threshold the actin cytoskeleton softens, decreasing loading rates and preventing reinforcement. By stretching rat lungs in vivo, we show that a similar phenomenon may occur. Our results show that cell sensing of external forces and of passive mechanical parameters (like tissue stiffness) can be understood through the same mechanisms, driven by the properties under force of the mechanosensing molecules involved.

[1] Institute for Bioengineering of Catalonia (IBEC), the Barcelona Institute of Technology (BIST), Barcelona, Spain. [2] Universitat de Barcelona, Barcelona, Spain. [3] Institute of Cell Biology, Center of Molecular Biology of Inflammation (ZMBE), University of Münster, Münster, Germany. [4] Universitat Politècnica de Catalunya (UPC), Campus Nord, Barcelona, Spain. [5] Department of Physics, King's College London, Strand, London, UK. [6] Harvard John A. Paulson School of Engineering and Applied Sciences, Harvard University, Cambridge, MA, USA. [7] Wyss Institute for Biologically Inspired Engineering, Boston, MA, USA. [8] Institució Catalana de Recerca i Estudis Avançats (ICREA), Passeig de Lluís Companys, Barcelona, Spain. [9] CIBER en Bioingeniería, Biomateriales y Nanomedicina (CIBER–BBN), Madrid, Spain. [10] CIBER de Enfermedades Respiratorias, Madrid, Spain. [11] Institut d'Investigacions Biomèdiques August Pi Sunyer, Barcelona, Spain. [12] These authors contributed equally: Ion Andreu, Bryan Falcones. ✉email: isaac.almendros@ub.edu; rocacusachs@ub.edu

Cells are constantly subjected to forces transmitted through tissues[1,2], which regulate major processes in health and disease[3,4]. Despite this importance, the fundamental mechanical variables that cells sense and respond to are not fully understood, and have been a matter of intense debate[5–7]. Mechanosensing molecules such as the integrin-actin adaptor protein talin respond to specific values of applied force[8], but it has been suggested that cells respond not directly to force but to the associated deformations exerted on the extracellular matrix (ECM)[5,9] or even to a combination of force and deformation[6,10]. However, in most scenarios forces and deformations are high dynamic[11–14], also because physiological ECMs are typically viscoelastic (and therefore have time-dependent responses)[15,16]. Thus, a sensing system based only on given fixed magnitudes of force or deformation may not be effective. Alternatively, cells could be sensitive to force dynamics per se. Specifically, force-sensitive molecular events such as bond rupture[17,18] or protein unfolding[19] have long been predicted and measured to depend on the rate of force application, known as the loading rate. This dependency is in fact an implicit underlying assumption of the molecular clutch theory, which has been employed to model how cells generate and transmit forces to sense passive mechanical factors such as ECM rigidity[20,21], viscosity[22], or ligand density[23]. In this framework, cells generate forces through actomyosin contraction, which results in a retrograde flow of actin from the cell edge to the cell center. As actin is connected to the ECM via integrins, this retrograde flow pulls on and deforms the substrate. ECM mechanics strongly influence the resulting loading rate, which is the product of the deformation speed times the effective stiffness of the substrate. Experimentally, this effect is for instance observed in the dynamics of cell pulling of micropillars of different stiffness[10]. Changes in loading rates then enable ECM mechanosensing by affecting molecular events such as integrin-ECM binding or talin unfolding[8,20]. This hypothesis is attractive in that it relates cell sensing of passive ECM mechanical factors to the sensing of directly applied forces, in a unified mechanism. It is also consistent with the well-known frequency dependence of cell mechanoresponse in many different systems[24–27]. However, whether the loading rate (rather than other static or dynamic mechanical variables) is a driving parameter of mechanosensing, is unknown.

In this work, we demonstrate the role of the loading rate in mechanosensing. First, we stretch cells at different frequencies and amplitudes, and show that their mechanosensing response (as measured by YAP nuclear localization and focal adhesion growth) can be explained by the applied loading rate, as predicted by a computational clutch model. The loading rate triggers two competing effects: talin-mediated adhesion growth and reinforcement (at intermediate rates), and the disruption and softening of the actin cytoskeleton (at high rates). Then, we verify that the loading rate triggers reinforcement at the sub-μm adhesion scale (using optical tweezers) and cytoskeletal softening at the cell scale (using Atomic Force Microscopy). Finally, we show in vivo that ventilating rat lungs at different frequencies also leads to effects on YAP nuclear localization.

## Results

### The cell stretch rate drives mechanosensing in a biphasic manner.
To start exploring the role of loading rate, we seeded mouse embryonic fibroblasts on very soft (0.6 kPa in rigidity) fibronectin-coated polyacrylamide gels. In these conditions, we measured actin retrograde flows via time-lapse imaging of lifeact-transfected cells, and mechanosensing via immunostaining of two well-known mechanosensitive features: paxillin-containing cell-matrix adhesions, and the nuclear translocation of the transcriptional regulator YAP[28–30]. Retrograde flows were of ~50 nm/s (Fig. 1a, b and Supplementary Video 1) and mechanosensing was not triggered, as cells showed largely cytosolic YAP and only small punctate paxillin-containing adhesions (Fig. 1c). This is consistent with previously reported phenotypes of cells on soft substrates, and is indicative of a fast flow caused by low adhesion (and thereby low actin attachment) to the ECM[20]. Consistently, actin flows of cells on stiff substrates but measured at the very edge of lamellipodia (where adhesion is still very low) were of a similar magnitude (Fig. 1a, b and Supplementary Video 2).

We then used two ways to increase the loading rate, considering that it is the product of deformation speed times substrate stiffness. First, we increased the loading rate indirectly, by increasing substrate stiffness. In this case, mechanosensing (as indicated by changes in YAP nuclear localization) was first observed when rigidity was increased over 5-fold to 3.4 kPa (Supplementary Fig. 1a). Second, we increased the loading rate directly, by externally applying stretch (a stimulus well-known to trigger mechanosensing[24–27]) to cells seeded on soft 0.6 kPa gels using a previously described device (Fig. 1d). From a given amplitude and frequency of applied stretch, we can estimate the corresponding cell deformation speeds. To this end, we consider the average spreading diameter of cells (~20 μm), and assume that cell–substrate attachment and force transmission occurs largely at the cell periphery, where focal adhesions were mostly located. Using this approach, applying a very mild stretch (2.5% biaxial stretch, applied cyclically with a triangular 0.125 Hz wave for 1 h) leads to a deformation speed of ~60 nm/s, of the same order of magnitude than internally generated actomyosin flows in low adhesion conditions. Consistently, this signal had only a very small effect on YAP or paxillin mechanosensing responses (Fig. 1e). However, when we increased deformation speeds by changing stretch frequency (and not amplitude), a clear response was observed above 1 Hz both for YAP (by progressively localizing in the nucleus) and for adhesions (by growing from punctate to the larger, elongated structures known as focal adhesions, Fig. 1e). Compared to the signal at 0.125 Hz, this represents an increase of 4–8-fold, again consistent with stiffness results. Whereas this equivalence is only approximate, and the magnitude of response was different in both cases (higher with stiffness than with stretch), the match in the order of magnitude support a role of the loading rate. Importantly, as previously reported for stiffness[20], the mechanosensing response to stretch was mediated by the mechanosensitive protein talin[20], since its knock-down eliminated both YAP and paxillin responses (Supplementary Fig. 1d–g).

We then repeated the frequency sweep by applying different amounts of stretch, from 2.5 to 20% (Fig. 1e–h, and Supplementary Fig. 2a–h). As expected from a role of loading rate, progressively increasing the stretch amplitude led to higher responses at most frequencies. However, for stretch amplitudes above 5%, increasing the frequency only increased response up to a point: above 1 Hz, fast stretching failed to trigger adhesion growth or YAP nuclear translocation (Fig. 1c, e–h). This was not due to cells detaching from the substrate, since spreading areas did not decrease (Supplementary Fig. 1b). To further verify the role of the stretch rate and to decouple it from that of stretch frequency, we stretched cells at a frequency of 1 Hz, but instead of applying a progressive triangular signal, we applied stretch as fast as possible. This led to a quasi-square signal where the stretch rate more than doubled with respect to the corresponding triangular signal, and was thereby somewhat above that of the triangular 2 Hz signal (Supplementary Fig. 2b, h). Accordingly and for all stretch amplitudes, applying a square rather than triangular 1 Hz signal led to similar results than those obtained at the 2 Hz triangular signal (Fig. 1c, e–h). Importantly, this observed

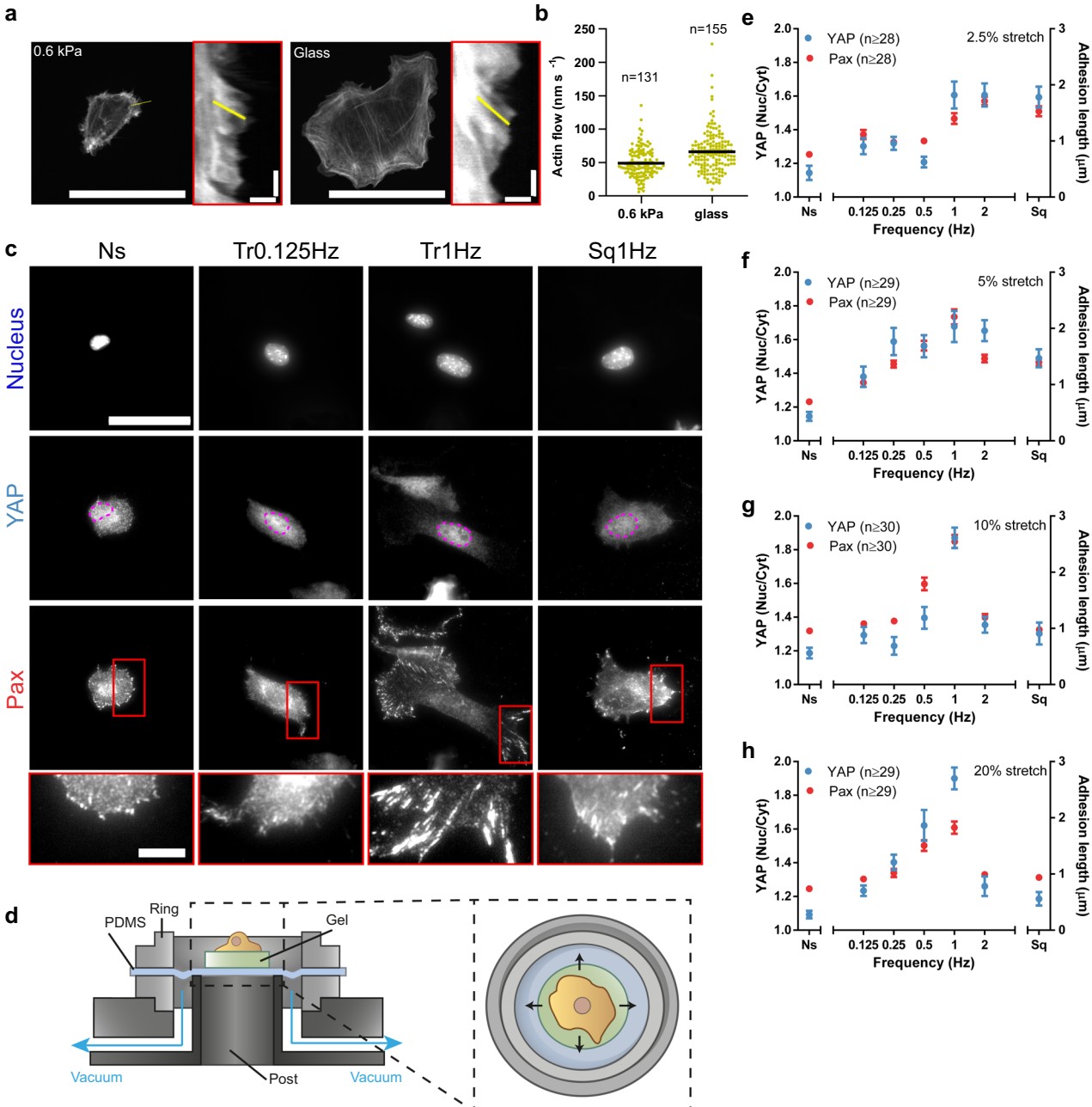

**Fig. 1 The rate of cell stretch drives mechanosensing in a biphasic manner. a** Cells transfected with LifeAct-GFP and plated on polyacrylamide gels of 0.6 kPa or on glass. Insets are kymographs showing the movement of actin features along the lines marked in yellow. **b** Actin retrograde flow of cells cultured on 0.6 kPa gels or glass. *n* numbers are traces. **c** Nuclear, YAP, and paxillin stainings of cells stretched by 10% using the setup using triangular (Tr) and square (Sq) signals at different frequencies. Ns non-stretched cells. In YAP images, magenta outlines indicate the nucleus. In paxillin images, areas circled in red are shown magnified below. **d** Illustration of the stretch setup. **e–h** Quantifications of YAP nuclear to cytoplasmic ratios and paxillin focal adhesion lengths for cells stretched at 2.5 % (**e**), 5% (**f**) 10% (**g**), and 20% (**h**). Results are shown for non-stretched cells (Ns), cells stretched with triangular signals at different frequencies, and cells stretched with a square signal at 1 Hz (Sq). The effects of frequency were significant for both YAP and paxillin in all panels (*p* < 0.0001). The effect of square versus triangular 1 Hz signals was significant for paxillin at 5% stretch (*p* = 0.0025) and for both YAP and paxillin for 10 and 20% stretch (*p* < 0.0001). Statistical significance was assessed with Kruskal–Wallis tests. *n* numbers are cells. Scale bars are 50 μm in cells, 2 μm/40 s in kymographs (*x/y* axes), and 10 μm in magnifications. Data are shown as mean ± s.e.m.

biphasic response was also observed in cell proliferation rates (Supplementary Fig. 1c), a well-known downstream effect of mechanosensing and YAP[28]. The biphasic response was also generalizable to other cell types (lung endothelial and epithelial cells, Supplementary Fig. 3).

To understand the decreased response at high frequencies, we hypothesized that it could be caused by the well-described

phenomenon by which both cells and actin gels soften when submitted to high stretch amplitudes[31–34] or rates[35]. This softening is caused by a partial disruption of the cytoskeleton, which is explained not by a single molecular event but by broad effects on the entire cytoskeletal network[31]. Of note, this phenomenon is often termed "fluidization"[33,34], which in precise terms involves both softening and an increased viscous-like

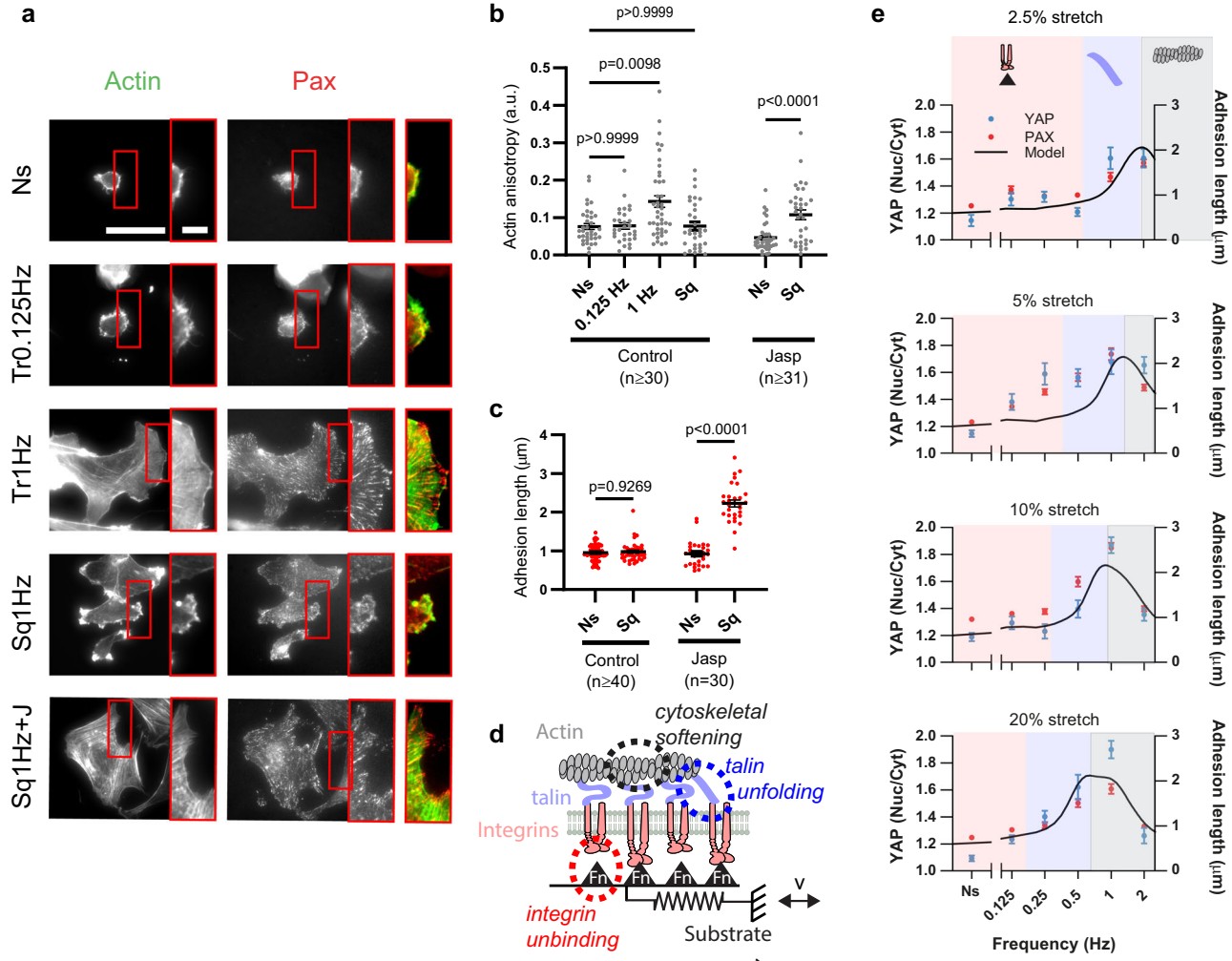

**Fig. 2 A molecular clutch model considering mechanosensing and cytoskeletal softening predicts the response to stretch. a** Actin and paxillin stainings for cells either not stretched or stretched by 10% with triangular signals (0.125 Hz, 1 Hz) or square signals (Sq, 1 Hz, with or without Jasplakinolide treatment). Areas circled in red are shown magnified at the right of each image, and shown as a merged image (actin, green, paxillin, red). Scale bar is 50 μm, and 10 μm in magnifications. **b**, **c** corresponding quantifications of actin anisotropy (**b**) and adhesion length (**c**) for control cells, and cells treated with Jasplakinolide. *n* numbers are cells. Statistical significance was assessed with two-way ANOVA. **d** Cartoon of computational clutch model (see "Methods" section). The model considers a relative speed of movement *v* between cell and substrate (given by stretch). The substrate is represented by an elastic spring with binding sites to integrins (via fibronectin, Fn triangles), which in turn connect to talin and actin. As stretch applies forces, these can lead to integrin unbinding, talin unfolding, or cytoskeletal softening. **e** Model predictions (black line) overlaid on experimental YAP and paxillin results from Fig. 1. The only parameter changing between simulations is stretch amplitude (1.5, 2, 2.5, and 3 μm for 2.5, 5, 10, and 20% stretch). Areas shaded in pink, blue, and gray show the regions dominated respectively by integrin unbinding, talin unfolding, and cytoskeletal softening. Data are shown as mean ± s.e.m.

behavior[31,36]. To assess this hypothesis, we evaluated cytoskeletal organization in the different conditions.

Cells stretched at very low or high rates (1 Hz square signal) showed a largely unstructured actin cytoskeleton (Fig. 2a). Accordingly, their actin anisotropy (a quantification of the degree of alignment and organization of actin fibers[37]) was low, at the level of unstretched cells (Fig. 2b). In contrast, cells stretched by 10% with a 1 Hz triangular signal, where focal adhesions were largest, formed clear actin stress fibers connected to adhesions (Fig. 2a), and exhibited high values of actin anisotropy (Fig. 2b). Then, we stretched cells treated with Jasplakinolide, a drug that stabilizes actin fibers by preventing their depolymerization[38]. Consistent with the role of cytoskeletal softening, Jasplakinolide treatment rescued the response to high stretch rates (1 Hz square signal), in terms of stress fiber formation (Fig. 2a), actin anisotropy (Fig. 2b), and focal adhesion formation (Fig. 2c).

**A clutch model considering how the loading rate affects reinforcement and cytoskeletal softening explains the results.** Our results can be explained by a role of loading rate, affecting both reinforcement and cytoskeletal softening. To assess how these different factors are related, we developed a computational clutch model based on our previous work[20]. This model considers progressive force application to links between actin, talin, integrins, and fibronectin (i.e., "clutches"). Then, it considers how talin unfolding and integrin-fibronectin unbinding depend on force (based on experimental single molecule data[39,40]). If talin unfolds before the clutch disengages from the substrate through integrin unbinding, we assume that there is a mechanosensing reinforcement event, which leads to integrin recruitment (i.e., adhesion growth in experiments). As a modification from our previous model, here we introduced that (i) force on clutches does not arise from actomyosin contractility, but from externally imposed periodic stretch, and (ii) the clutch can be disengaged not only by

integrin unbinding, but also by actin cytoskeleton disruption (i.e., softening) above a threshold force. Importantly, the model does not assume any dependence on the loading rate per se. Rather, increasing loading rates simply reduce the amount of time required to reach a given force level, thereby increasing the likelihood that high forces will be reached before clutch disengagement. Thus, different loading rates mean that different force regimes are reached, which may differently affect the molecular events considered (integrin unbinding, talin unfolding, or cytoskeletal softening).

By modifying only the parameters of applied stretch frequency and amplitude, the model largely reproduced observed experimental trends (Fig. 2e). For low frequencies, low loading rates mean that forces stayed in a regime where integrin unbinding occurred first, preventing mechanosensing. At higher frequencies, higher forces were reached, progressively allowing talin unfolding, and increasing mechanosensing. However, at very high frequencies, the very high forces required for cytoskeletal softening were reached. By increasing stretch amplitude, loading rates are achieved at lower frequencies, shifting the curves to the left, and effectively making the softening regime observable only for high amplitudes. Of note, the cytoskeletal softening event in the model cannot distinguish between different potential events, such as breaking of actin filaments, or severing of actin crosslinks, for instance. However, the model provided a good fit to the data by assuming a force of about 140 pN, in reasonable agreement with reported experimental values for the breaking of actin filaments[41].

**The loading rate drives the maturation of single adhesions**. Our results and modeling are consistent with a role of loading rate, which would trigger mechanosensing or cytoskeletal softening depending on its magnitude. To further verify this, we tested some assumptions of our hypothesis and model which could not be addressed through the stretch device. First, effects should be caused by the loading rate and not the cell deformation rate, which also increased with stretch. Second, the effects are expected to occur not only at the global cell scale, but also at the local adhesion scale. To address these questions, we used a previously described optical tweezers setup[42] (Fig. 3a). We seeded cells transfected with GFP-paxillin on glass, trapped fibronectin-coated 1 μm diameter beads, attached them to the cell surface, and applied forces to cells by displacing the optical trap horizontally with triangular signals of the same amplitude, but different frequencies (Fig. 3b, c and Supplementary Fig. 4a). Initially, this stimulation led to bead displacements of ~0.2 μm and applied forces of 10–15 pN, which did not show any significant trend with frequency (Supplementary Fig. 4b–d). With time, force application led to the mechanosensing process known as adhesion reinforcement (Fig. 3b). This was characterized by a progressive reduction in bead displacements (Fig. 3c) and speeds (Fig. 3d), a measure of applied deformation rates. Concomitantly, there was an increase in applied forces (Fig. 3c), loading rates (Fig. 3e), the effective stiffness of beads (the ratio between forces and displacements, Fig. 3f), and recruitment of paxillin to beads (Fig. 3b, g). As previously described[43], unstimulated beads did not recruit paxillin (Fig. 3b, k). High frequencies led to higher deformation rates (speed), loading rates, bead stiffness, and paxillin recruitment than low frequencies (Fig. 3h–k). However, and unlike in the case of stretch, the response was monotonic, and no decrease in reinforcement or paxillin recruitment was observed even at very high frequencies (Fig. 3k). Applying a square rather than triangular 1 Hz signal markedly increased force loading rates by almost two orders of magnitude (Fig. 3i). Accordingly, it increased, rather than decreased, paxillin recruitment (Fig. 3k).

These results demonstrate that effects happen at the local adhesion scale. Further, they are consistent with a role of the loading rate, which, unlike the deformation rate, increased concomitantly with paxillin recruitment both with time (within each experiment) and with frequency. Of note, the absence of the cytoskeletal softening regime in these experiments is also to be expected, given that optical tweezers cannot apply forces larger than ~100 pN. This is consistent with the observation that cytoskeletal stiffness increased (rather than decreased) with frequency when similarly small forces were applied with magnetic twisting cytometry[44,45]. Additionally, optical tweezers experiments were carried out in the lamellar region of cells seeded on stiff substrates. This region exhibited stress fibers, and therefore an actin network much more structured than on the rounded cell phenotype found on soft substrates before stretch (Supplementary Fig. 3). Thus, forces applied to beads are likely distributed among many filaments, reducing the likelihood of cytoskeletal softening.

**Cytoskeletal softening limits mechanosensing at high deformation rates**. Our data are consistent with cytoskeletal softening happening in the case of stretch experiments, but not optical tweezers. To test this experimentally, we attached cells in suspension to a flat Atomic Force Microscope (AFM) cantilever, placed them in contact with a fibronectin-coated glass, and pulled at different speeds (Fig. 4a–c). In these conditions, cells remained rounded, thereby mimicking the low-stiffness phenotype that cells exhibited before being stretched. We then measured the effective stiffness (Young's modulus) of cells as they were being pulled and thereby stretched (Fig. 4d). Increasing pulling speeds first increased stiffness, as typically occurs in cells or cytoskeletal networks[44–48]. However, between pulling speeds of 5 and 6 μm/s there was a sharp decrease, indicating a partial cytoskeletal disruption, or softening. Calculating an approximate equivalence, the range of stretch at which cells failed to mechanosense (10–20% stretch between 1 and 2 Hz, for cells of ~20 μm in size) corresponds to 2–8 μm/s in deformation, thereby matching the order of magnitude of AFM results (see also Supplementary Table 1). Interestingly, this decrease in stiffness was also associated with lower cell detachment forces, as previously reported[49] (Supplementary fig. 5a). To mimic optical tweezers experiments, we carried out a modified experiment in which we attached fibronectin-coated 3 μm beads to AFM cantilevers, placed them in contact with the surface of previously adhered fibroblasts, and pulled at different speeds (Fig. 4e–g). Consistently with optical tweezers experiments, no softening was observed (Fig. 4h). Of note, measured stiffness values in AFM bead experiments were at least one order of magnitude higher than in whole cell AFM experiments, confirming the notion that cellular lamella has a more structured actin cytoskeleton than rounded cells (Fig. 4d, h).

Finally, we tested another interesting implication of our hypothesis. Loading rates depend not only on the rate of deformation but also on the stiffness of the structure being deformed (which will also impact the amplitude of loads that can be reached). In stretch experiments, the deforming structure is the entire cell, and thus the ability to form stress fibers spanning the cell body (which increases cell stiffness[50]) should be crucial to enable mechanosensing responses. In contrast, in optical tweezers experiments, the presence of a local actin meshwork should be sufficient. To test this, we treated cells with the myosin contractility inhibitor blebbistatin[51]. Although myosin inhibition may impact cell mechanochemical signaling in complex ways[52], a clear effect was that actin stress fibers were eliminated, but a dendritic actin network was retained in the lamellar region (Supplementary Fig. 5c, d). In such conditions and as expected,

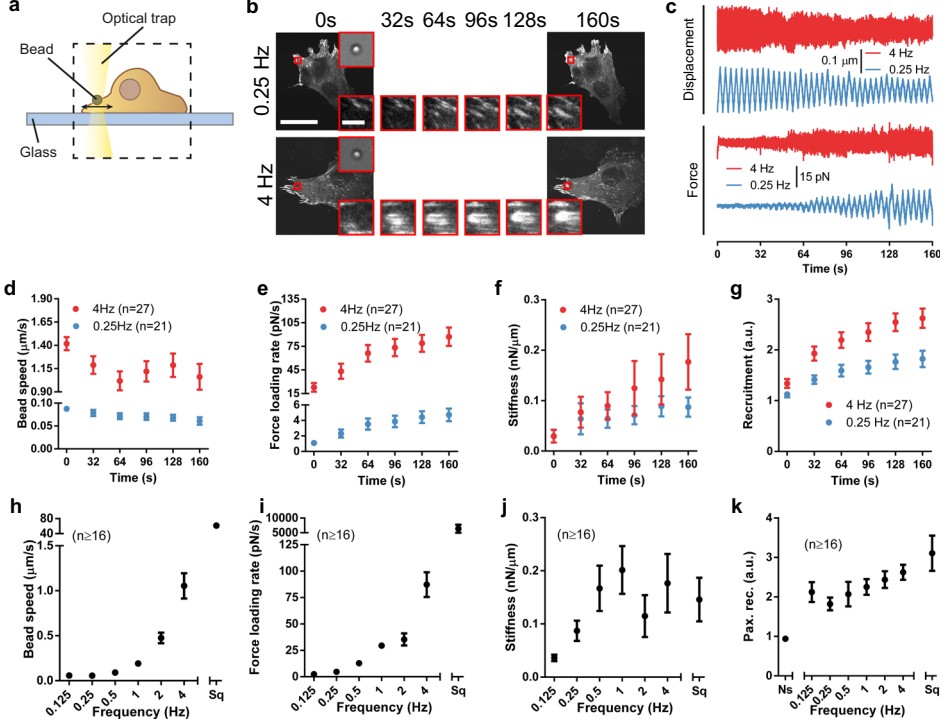

**Fig. 3 The loading rate of force application to single adhesions drives their maturation. a** Illustration of the optical tweezer setup. **b** Images of cells transfected with GFP-paxillin during force application with triangular signals at 0.25 and 4 Hz, shown as a function of time. The area circled in red indicates the position of the stimulated bead, which is shown magnified at the top-right corner (brightfield image) and bottom-right corner (GFP-paxillin image). Magnified GFP-paxillin images are shown at different timepoints. **c** Example traces of displacement and forces for beads stimulated at 4 and 0.25 Hz. **d–g** Bead speed (**d**), force loading rate (**e**), stiffness (**f**), and recruitment of GFP-paxillin at beads (**g**) as a function of time for beads stimulated at 4 Hz and 0.25 Hz. **h–k** Bead speed (**h**), force loading rate (**i**), stiffness (**j**), and recruitment of GFP-paxillin at beads (**k**) for beads at the end of the experiment (160 s) for all conditions. The effects of frequency on both stiffness and paxillin recruitment were significant ($p = 0.002$ and 0.008, respectively). Ns non-stimulated beads, Sq stimulation with a 1 Hz square signal. $n$ numbers are beads in all panels. Statistical significance was assessed with Kruskal–Wallis test. Scale bar is 50 μm, and 5 μm in magnifications. Data are shown as mean ± s.e.m.

mechanosensing was abrogated in response to stretch (Fig. 4i–l), but not to local force application (Fig. 4m–o).

**Increasing rates of lung ventilation in vivo induce YAP nuclear localization.** To conclude our study, we assessed whether the role of the force loading rate could also be observed at an organ level in vivo. To this end, we used a previously described setup[53] to subject each of the two lungs of a rat to independent mechanical ventilation (Fig. 5a). This setting allowed us to increase ventilation frequency in one lung while decreasing it in the other, in such a way that total ventilation was kept constant. Through this approach, we were able to compare the effects of locally varying the mechanical loading rate in lungs without interfering with systemic animal gas exchange. After 1 h of stimulation, lungs were excised and immunostained, and lung alveoli (containing mostly endothelial and epithelial cells)[54] were imaged. The 3D, in vivo setup led to paxillin immunostainings without sufficient resolution to quantify adhesion shapes, but YAP nuclear to cytosolic ratios could be assessed (Fig. 5b, c and Supplementary Fig. 6). Whereas, as expected, no significant differences were found between right and left lungs when ventilated at the same frequency (Fig. 5d), we found increased levels of nuclear YAP with increased ventilation frequency (Fig. 5b, c and Supplementary Fig. 5). These results could not be explained by regulation of oxygen or $CO_2$ levels, since overall ventilation in the animal was kept constant when differential ventilation was applied to both lungs, and cells in the lung parenchyma are perfused with systemic oxygenated blood[55,56]. Further and although there are no specific studies in lungs, hypoxia consistently increases YAP levels

in several tissues[57–59], as opposed to the effect we see in slowly ventilated (0.1 Hz) lungs. Live tissues are highly complex environments where neither force transmission nor ensuing signaling can be precisely controlled, and the precise mechanisms by which mechanical signals, oxygen levels, and other factors regulate YAP signaling in lungs remain to be elucidated. However, our results are consistent with a role of loading rate in controlling ventilation-induced mechanotransduction at the organ level in the lungs.

## Discussion

In summary, our results show that force loading rates drive mechanosensing by increasing reinforcement and adhesion growth at the local adhesion level, in a talin-dependent way and as predicted by a molecular clutch model. The range of relevant loading rates at the molecular level is hard to assess, but loading rate dependencies have been predicted to apply over extremely wide ranges[17], as confirmed for instance in integrin-ECM bonds ($10^1 - 10^5$ pN/s)[18]. If one assumes a density of bound integrins from $10^0$ to $10^2$ per μm² (from initial to stable adhesions[60,61]), our range of applied loading rates (Supplementary Table 1) would be largely expected to fall within this range. If loading rates are too high, the cytoskeleton undergoes softening, impairing mechanosensing. This softening event most likely involves the disruption of actin filaments, since it was prevented by a drug that stabilizes them (Jasplakinolide)[38]. Although not explored here, we note that other cytoskeletal components may also be involved in this softening response. Particularly, intermediate filaments (vimentin and keratins) play a major role in withstanding stress

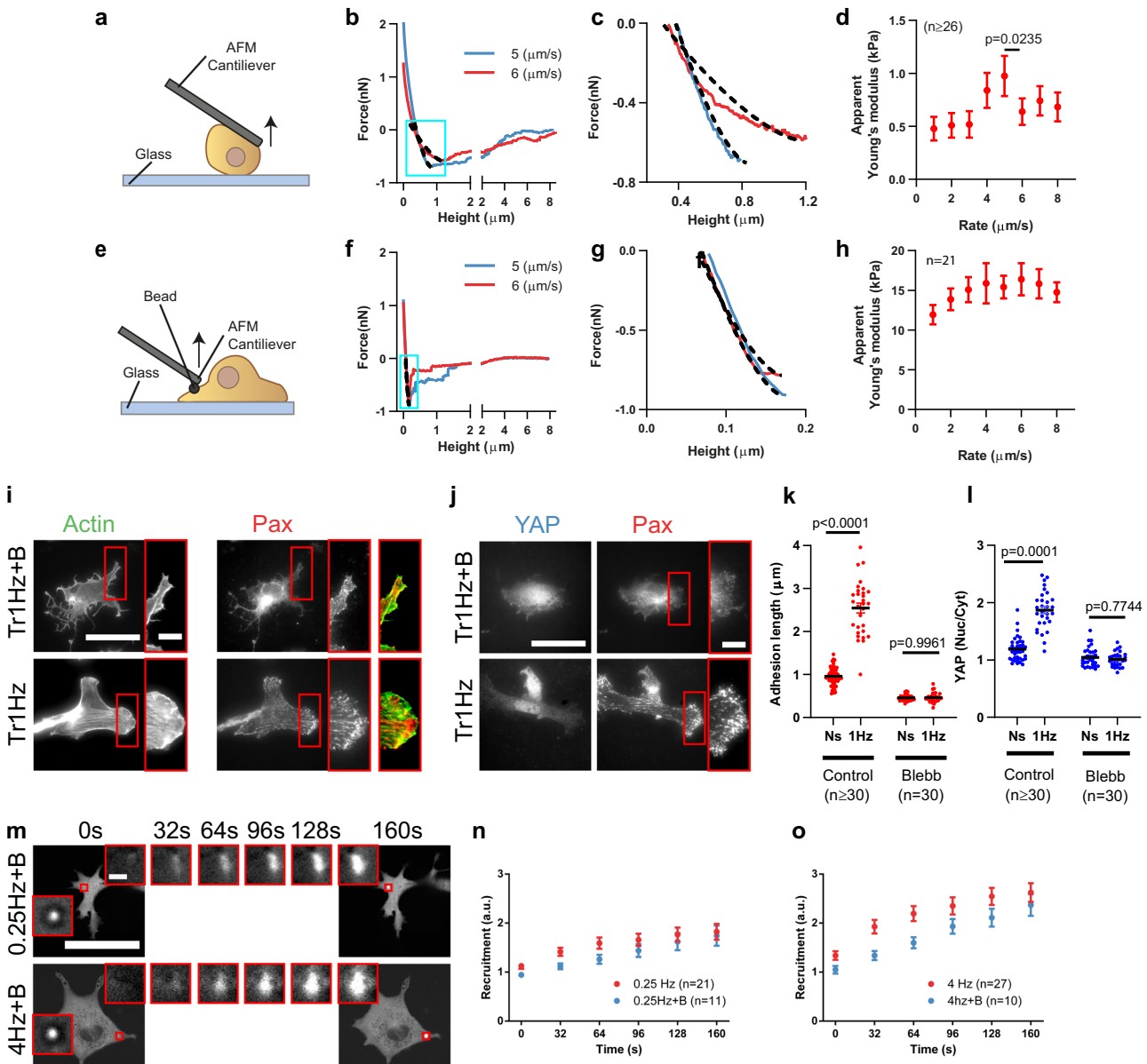

**Fig. 4 High deformation rates lead to cytoskeletal softening. a** Illustration of the single-cell AFM setup. **b** Example cantilever retraction curves of the single-cell AFM experiments. Dashed lines within blue squares show the fits of the force/deformation curves used to calculate apparent cell stiffness (Young's modulus). Curves were fitted between the beginning of cell stretch (below zero force) and the point at which cells start detaching (minimum force in the curve). **c** Magnification of the curve section fitted in **b**. **d** Stiffness as a function of the retraction speed for cells attaching to a fibronectin-coated substrate. The effect of retraction speed ($p < 0.0001$), and the specific decrease from 5 to 6 µm/s ($p = 0.0235$) were significant. $n$ numbers are curves. Statistical significance was assessed with Friedman test. **e** Illustration of the bead AFM setup. **f** Example cantilever retraction curves of bead AFM experiments. **g** Magnification of the curve section fitted in **f**. **h** Stiffness as a function of the retraction speed for fibronectin-coated beads attaching to cells. The effect of retraction speed was not significant (Friedman test). $n$ numbers are curves. **i**, **j** Actin and paxillin (**i**) and YAP and paxillin (**j**) stainings for cells stretched by 10% with a triangular 1 Hz signal, with or without blebbistatin treatment. Scale bar is 50 µm (10 µm in magnifications). **k**, **l** Corresponding quantifications of adhesion length (**k**) and YAP nuclear to cytosolic ratios (**l**). $n$ numbers are cells. Statistical significance was assessed with two-way ANOVA. **m** Images of blebbistatin-treated cells transfected with GFP-paxillin during force application with triangular signals at 0.25 and 4 Hz, shown as a function of time. Areas circled in red are shown magnified at different timepoints. Scale bar is 50 µm (2 µm in magnifications). **n**, **o** Corresponding quantifications of recruitment of GFP-paxillin at beads at 0.25 Hz (**n**) and 4 Hz (**o**). No significant effect of blebbistatin was observed (two-way ANOVA). $n$ numbers are beads. Data are shown as mean ± s.e.m.

under high loads[34,62,63], and may have an important role in regulating (or even buffering) softening responses in different scenarios.

Our work provides a unifying mechanism to understand how cells respond not only to directly applied forces, but also to passive mechanical stimuli such as tissue rigidity or ECM ligand distribution, where we have reported similar biphasic

dependencies of focal adhesions and YAP localization[23]. Further, it also provides a framework to understand how the seemingly opposed concepts of reinforcement and cytoskeletal softening, previously analyzed within the context of cell rheology[32,64], are coupled to drive mechanosensing. Potentially, this framework could be extended to explain mechanosensing mechanisms beyond focal adhesion formation and YAP, such as the actin-

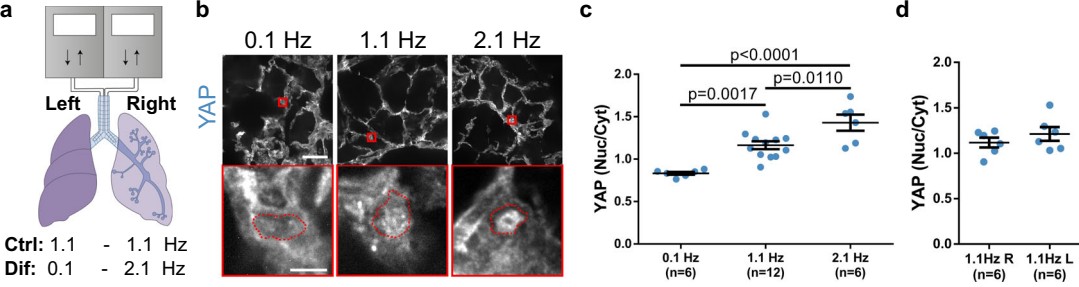

**Fig. 5 Increasing rates of lung ventilation in vivo induce YAP nuclear localization. a** Diagram of the rat lung ventilation setup, where each lung was independently cannulated and ventilated. Ventilation was either uniform at 1.1 Hz in both lungs (Ctrl) or differential with the left and right lungs ventilated at 0.1 and 2.1 Hz, respectively (Dif). Both conditions had the same ventilation volume. **b** YAP staining of rat lungs ventilated at 0.1, 1.1, and 2.1 Hz. Areas circled in red are magnified below each image. In magnified images, nuclear contours (as determined from Hoechst stainings) are shown in red. Scale bar is 50 μm in top images, and 5 μm in magnifications. **c** Quantification of YAP nuclear to cytoplasmic ratios for rat lungs ventilated with same tidal volume at 0.1, 1.1, and 2.1 Hz. Statistical significance was assessed with one-way ANOVA. **d** Quantification of YAP nuclear to cytoplasmic ratios for left and right rat lungs ventilated at 1.1 Hz. *n* numbers are rat lungs. No significant differences were observed (two-sided *t*-test). Data are shown as mean ± s.e.m.

dependent nuclear localization of MRTF-A[65]. In vivo, the extremely wide range of loading rates in different contexts (from very fast in the respiratory[2] or cardiovascular[1] systems, or in vocal cord vibration[66], to very slow in progressive ECM remodeling in cancer[3]), could thus be central to understand how mechanosensing is regulated. In lung alveoli, we only observed the initial reinforcement phase, even though relevant cell types (fibroblasts, endothelial cells, and epithelial cells) all exhibited both phases in vitro. Whereas the mechanisms remain to be studied, this suggests potential large-scale mechanisms at the tissue level to buffer against cytoskeletal damage. However, in other contexts, both reinforcement and softening may be at place, and could even be harnessed to establish the levels of mechanical loading required to trigger specific responses.

## Methods

**Cell culture and reagents**. Mouse embryonic fibroblasts (MEFs) were cultured as previously described[67], using Dulbecco's modified eagle medium (DMEM, ThermoFisher Scientific, 41965-039) supplemented with 10% FBS (ThermoFisher Scientific, 10270-106) and 1% penicillin–streptomycin (ThermoFisher Scientific, 10378-016), and 1.5% HEPES 1M (Sigma Aldrich, H0887). Talin 1$^{-/-}$ MEFs were cultured as previously described[68], using DMEM supplemented with 15% FBS, 1% penicillin–streptomycin, and 1.5% HEPES 1 M. Primary human small airway epithelial cells (SAEC, ATCC® PCS-301-010™) were purchased from ATCC, cultured in Airway Epithelial Cell Basal Medium (ATCC® PCS-300-030™) supplemented with Bronchial Epithelial Cell Growth Kit (ATCC® PCS-300-040™) and 1% penicillin–streptomycin. Primary human lung microvascular endothelial cells (HMVEC, CC-2527) were purchased from Lonza,cultured using Vascular Cell Basal Medium (ATCC® PCS-100-030), supplemented with Microvascular Endothelial Cell Growth Kit-VEGF (ATCC® PCS-110-041) and 12.5 μg/mL blasticidin. Cell cultures were routinely checked for mycoplasma. $CO_2$-independent media was prepared by using $CO_2$-independent DMEM (ThermoFisher Scientific, 18045 -054) supplemented with 10% FBS, 1% penicillin–streptomycin, 1.5% HEPES 1 M, and 2% L-Glutamine (ThermoFisher Scientific, 25030-024). Media for optical tweezers experiments was supplemented with Rutin (ThermoFisher Scientific, 132391000) 10 mg/L right before the experiment.

**Transfection**. Talin 2 was knocked down as previously described[68], by transfecting Talin 1$^{-/-}$ MEFs using the Neon Transfection System (ThermoFisher Scientific) with a plasmid encoding a short hairpin RNA (shRNA) targeting the nucleotide sequence 5′-GATCCGAAGTCAGTATTACGTTGTTCTCAAGAGAAA-CAACGTAATACTGACTTCTTTTTTTCTAGAG-3′. For retrograde flow and optical tweezers experiments, cells were transfected with either lifeact-GFP[20] or pEGFP-Paxillin[69], using the Nucleofactor 2b Device (Lonza).

**Antibodies and compounds**. Primary antibodies used were anti-Paxillin rabbit clonal (Y113, Abcam, ab32084, 1:200 dilution), and anti-YAP mouse monoclonal (63.7, Santa Cruz Biotechnology, sc-101199, 1:200 dilution). Secondary antibodies used were Alexa Fluor 488 anti-mouse (A-11029, ThermoFisher Scientific), Alexa Fluor 488 anti-rabbit (A-21206; ThermoFisher Scientific), and Alexa Fluor 555 anti-rabbit (A-21429, ThermoFisher Scientific), all at 1:500 dilution. Compounds used were Blebbistatin (Sigma Aldrich, 50 μM), Jasplakinolide (J4580, Sigma

Aldrich, 25 nM), phalloidin (Alexa Fluor 555 phalloidin, ThermoFisher Scientific, 1:1000 dilution), and Hoechst (33342, ThermoFisher Scientific, 1:2000 dilution).

**Retrograde flow experiments**. Cells were transfected with LifeAct-GFP, plated on gels or glass coated with 10 μg/mL fibronectin, and imaged every 2 s for 4 min with a ×60 water immersion objective (NA 1.20, Nikon) and a spinning-disc confocal microscope (Andor). For each cell, kymographs were obtained at the cell periphery, and actin speed was measured from the slope of actin features observed in the kymographs.

**Preparation of stretchable membranes**. Stretchable polydimethylsiloxane (Sylgard 184 Silicone Elastomer Kit, Dow Corning) membranes were prepared as previously described[70]. A mix of 10:1 base to crosslinker ratio was spun for 1 min at 500 rpm and cured at 65 °C overnight on plastic supports. Once polymerized, membranes were peeled off and assembled onto the stretching device. After assembly, membranes were plasma cleaned for 1 min, treated with 3-aminopropyl triethoxysilane (APTES, Sigma Aldrich) 10% in ethanol for 1 h at 65 °C, and with glutaraldehyde (Sigma Aldrich) 1.5% in phosphate-buffered saline 1× (PBS, Sigma Aldrich) for 25 min at room temperature.

Then, polyacrylamide gels were prepared and attached to membranes. To this end, polyacrylamide gels were first prepared by adapting previous protocols[23,71]. Polyacrylamide gels were polymerized between two glass coverslips treated with 2% dimethyldichlorosilane (Plus One Repel Silane, GE Healthcare). For 0.6 kPa gels, the mix contained 4% acrylamide (BioRad), 0.03% BisAcrylamide (BioRad), 2% 200-nm-diameter dark red fluorescence carboxylate-modified beads (Fluospheres, ThermoFisher Scientific), 0.5% ammonium persulphate (APS, Sigma Aldrich), and 0.05% tetramethylethylenediamine (TEMED, Sigma Aldrich), in PBS 1×. After polymerization, one coverslip was detached, and the gel was then attached to the PDMS membrane. To this end, it was pressed against the prepared stretchable PDMS membrane and left overnight at 37 °C in an incubator with humidity control. The remaining coverslip was removed the next day.

Polyacrylamide gels were coated using a protocol adapted from the literature[72]. Briefly, gels were covered with a mix containing 10% HEPES 0.5 M pH 6, 0.002% BisAcrylamide (BioRad), 0.3% 10 mg/mL N-hydroxysuccinimide (NHS, Sigma Aldrich) in dimethyl sulfoxide (DMSO, Sigma Aldrich), 1% Igracure (Sigma Aldrich), 0.0012% tetraacrylate (Sigma Aldrich), in Milli-Q water. Gels were then covered with a glass coverslip and illuminated with UV light for 10 min. After exposure, the glass coverslip was removed, and gels were washed twice with HEPES 25 mM pH 6 6 and twice again with PBS. Gels were then incubated with 10 μg/mL of fibronectin in PBS overnight at 8 °C, washed the next day thrice with PBS, and immediately used. The rigidity of the gels was measured using Atomic Force Microscopy as previously described[70] (Supplementary Table 2).

**Cell stretch**. Cells were seeded on 0.6 kPa gels attached to previously mounted stretchable PDMS membranes. After attachment, cell media was changed to $CO_2$-independent media. Cells were then stretched at 37 °C continuously for 1 h with one signal type (square or triangular), at one amplitude (20%, 10%, 5%, or 2.5%), and at one frequency (2 Hz, 1 Hz, 0.5 Hz, 0.25 Hz, 0.125 Hz) to produce different force loading rates. After stimulation, cells were immediately fixed and prepared for immunostaining. To study cell proliferation, cells were stretched using the same protocol for 2.5 h, and then the Click-iT™ EdU Cell Proliferation Kit (Invitrogen) was used according to manufacturer instructions.

**Immunostainings**. Immunostainings were performed as previously described[23]. Cells were fixed with 4% paraformaldehyde for 10 min, permeabilized with 0.1% Triton X-100 for 4 min, blocked with 2% Fish-Gelatin in PBS 1X for 1 h, incubated

with primary antibody for 1 h, washed with Fish-Gelatin-PBS for 30 min, incubated with secondary antibody for 1 h, washed with Fish-Gelatin-PBS for 30 min, and mounted using ProLong Gold Antifade Mountant (ThermoFisher Scientific).

For immunostainings of animal tissue, at the end-point of experiments animals were sacrificed by exsanguination and during residual heart beating lungs were perfused through the vasculature with ice-cold PBS 1×. Lungs were immediately excised en bloc with the heart and intrabronchial cannulas and perfused with cold OCT:PBS (3:1) (Optimal Cutting compound, Company). Lungs were then placed in cassettes with OCT on a dry ice platform and frozen at −80 °C. Lung blocks 70-µm thick were cut with a cryostat (ThermoScientific, Massachusetts, USA) and attached to slides. Lung slices were then washed with PBS and fixed with paraformaldehyde 4%. and After three more washes, tissue was permeabilized with 0.2 % Triton X-100, blocked with 10% FBS, and incubated for 30 s with TrueBlack (Biotium) to reduce ECM autofluorescence. The primary antibody was incubated overnight at 4 °C, and after three washes the secondary antibody was incubated for 90 min at room temperature. Finally, lung slices were counterstained with NucBlue (ThermoScientific, Massachusetts, USA) to stain the nuclei and mounted with Fluoromount (Dako). Two lung slices from each lung were imaged in three different fields to a total of six images per condition.

Once the samples were prepared, images of stretched cells were acquired with ×60 objective (NIR Apo 60X/WD 2.8, Nikon) with an upright microscope, images of cells on glass were acquired with ×60 objective (Plan Apo VC 60X/WD 0.31–0.28, Nikon) with a confocal inverted microscope, and images of animal tissue were acquired with ×60 objective (Plan Apo VC Oil 60X/WD 0.13, Nikon) with a confocal inverted microscope. Metamorph or Micromanager[73] software was used to control the microscopes.

**Image analysis**. Focal adhesion length was quantified manually by assessing the length of three representative adhesions in paxillin stainings at the cell edge, for n cells. Nuclear to cytoplasmic ratio of YAP was quantified manually by segmenting the nucleus using Hoechst (single cells) or NucBlue (rat lung slices) and using the following formula:

$$\text{a.u.} = \frac{I_{nucleus} - I_{background}}{I_{cytoplasm} - I_{background}} \quad (1)$$

Similarly, protein recruitment to beads was quantified manually by segmenting the bead area using the brightfield image and using the following formula:

$$\text{a.u.} = \frac{I_{bead} - I_{background}}{I_{cytoplasm} - I_{background}} \quad (2)$$

In both cases, $I_{cytoplasm}$ and $I_{background}$ refer to the average fluorescence intensity of the cytoplasm and background (i.e., areas with no cells). $I_{bead}$ and $I_{nucleus}$ refer to the average fluorescence intensity of the nucleus and bead. In the case of rat lung immunostainings, lung cuts were randomized before quantification. Six (6) areas coming from two (2) different cuts were analyzed, where twenty (20) cells per area were quantified taking the closest from the geometric center of the image.

The anisotropy of the actin cytoskeleton was determined by first manually segmenting the cells, followed by analysis using the ImageJ plug-in, FibrilTool[37].

**Optical tweezers**. The optical tweezers system was adapted from a previous setup[42]. Briefly, the optical tweezer uses a near-infrared fiber laser ($\lambda = 1064$ nm, YLR-5-LP; IPG Photonics) that passes through two acoustooptical modulators (DTSX-400-1064; AA Opto-Electronic) which are modulated by a variable frequency driver (Voltage Controlled Oscillator, DRFA10Y2X-D41k-34-50.110; AA Opto-Electronic). After modulation, the beam size is expanded and coupled into an inverted microscope (Eclipse Ti-e; Nikon Corporation) from the rear port. The beam is coupled in the optical path of the microscope by a dichroic mirror and focused in the object plane through a water immersion objective (Plan Apo VC WI, ×60, NA = 1.2; Nikon Corporation). To measure the force, the condenser was replaced by a force sensor module (Lunam T-40i; Impetux Optics, S.L.), which was positioned according to the manufacturer's procedure. The module is precalibrated and gives direct access to the force applied by the tweezer on any trapped object. To correct for focal drifts during the measurements, the Perfect Focus System (PFS, Nikon Corporation) was used. The sample is heated by a self-built heating chamber, keeping it at 37 °C. Image acquisition was done using a spinning disk system (CSU-W1 (Yokogawa); Intelligent Imaging Innovations Inc.) and a CMOS camera (Orca-flash4.0v2; Hamamatsu Photonics K.K.). All hardware was controlled using the custom-written LabVIEW programs (National Instruments Corporation).

Beads were coated as described previously[74]. Briefly, carboxylated 1 µm polystyrene beads (Micromod) were coated with a mixture of biotinylated pentameric FN7–10 (a four-domain segment of fibronectin responsible for cell binding and containing the RGD and PHSRN motifs[75]) and biotinylated bovine serum albumin at a ratio of 1:10.

Glass slides were coated with fibronectin 10 µg/mL in PBS overnight and rinsed thrice with PBS. Cells were then seeded on the glass. Once attached, beads were added. One bead was subsequently trapped and placed on the surface of a cell, and the stimulation was then started by using 120 mW of laser power to displace the bead 0.1 µm in x and y from the center of the trap for 160 s by using a triangle or

square signal of one frequency (4 Hz, 2 Hz, 1 Hz, 0.5 Hz, 0.25 Hz, 0.125 Hz) to produce different force loading rates. To compensate for the cell dragging the bead toward the nucleus, the optical trap was repositioned at intervals of 32 s.

Bead speeds and force loading rates were measured using respectively the displacement and force signals of the optical trap. Using a custom-made MATLAB program, each signal was detrended and then divided into linear segments of individual cycles and fitted to straight lines to obtain the slopes. The stiffness was calculated as described previously[74] by estimating the transfer function between the force and the displacement data at intervals of 32 s, and at the frequency of stimulation.

**Atomic force microscopy**. AFM experiments were carried out in a Nanowizard 4 AFM (JPK) mounted on top of a Nikon Ti Eclipse microscope. For experiments pulling on beads, Fibronectin or biotin-BSA coated beads were functionalized as described previously[74]. Briefly, carboxylated 3 µm silica beads (Polysciences) were coated with a mixture of biotinylated pentameric FN7–10 (a four-domain segment of fibronectin responsible for cell binding and containing the RGD and PHSRN motifs[75]) and biotinylated bovine serum albumin (Sigma Aldrich) at a ratio of 1:10. The beads were then attached to the cantilevers using a non-fluorescent adhesive (NOA63, Norland Products) to the end of tipless MLCT cantilevers (Veeco). Cells were seeded on fibronectin-coated coverslips and a force curve at each retraction velocity was acquired for each of the cells. Cells were kept at 37 °C using a BioCell (JPK). The spring constant of the cantilevers was calibrated by thermal tuning using the simple harmonic oscillator model.

For experiments pulling on cells, we followed the protocol described in the literature[49]. Briefly, cantilevers were submerged in sulfuric acid 1 M for 1 h. They were then washed with Milli-Q water and plasma cleaned for three minutes. After this, cantilevers were incubated with 0.5 mg/mL biotin-BSA and left overnight in a humid chamber at 37 °C. Next day they were washed thrice with PBS and incubated with 0.5 mg/mL Streptavidin and left in a humid chamber at room temperature for 30 min. Again, they were washed thrice with PBS and reincubated with 0.4 mg/mL biotin-ConcanavalinA for 30 min in a humid chamber at room temperature, after which they were washed thrice with PBS and stored under PBS for use. We used a Nanowizard 4 AFM (JPK) on top of a Nikon Ti Eclipse microscope. ConcanavalinA-coated MLCT-O cantilevers were calibrated by thermal tuning. Then, cells were trypsinized, resuspended in $CO_2$ independent media, and allowed to recover for 5 min. Rounded cells were attached to MLCT cantilevers by exerting 3 nN forces on top of a region of a coverslip with no coating and incubated for 5 min. The measurement of the adhesion forces was done by approaching the cell at the same velocity as the corresponding withdrawal velocity, keeping for 10 s and withdrawing until the cell was fully detached.

The maximum detachment force was determined by analyzing the retraction F–D curve. A MATLAB program was used to extract and analyze force-displacement curves from JPK data files. Forces were corrected as described in the literature[76] by subtracting the baseline offset, i.e., the value obtained at the end of curves when cells were fully detached. We calculated apparent stiffness (Young's modulus) by fitting retraction curves between the values of zero force and the maximum detachment force $F_{ad}$, defined as the absolute value of the minimum of the curve. This corresponds to the part of the curve from the onset of cell stretching (positive force values at the beginning of curves correspond to cells still in compression) to the point at which cells start detaching. Curves were fitted to the Derjagin, Muller, Toropov (DMT)[77,78] contact model, which considers the force-indentation relationship between a flat surface and a sphere, taking into account both elastic deformations and adhesion between the surfaces. Specifically, we fitted the curve:

$$F + F_{ad} = ER^{\frac{1}{2}}|\delta - \delta_{ad}|^{\frac{3}{2}} \quad (3)$$

Where $F$ is force, $E$ is Young's modulus, $R$ is the average radius of cells or beads depending on the experiment (which we took as 10 and 1.5 µm, respectively), $\delta$ is indentation, and $\delta_{ad}$ is the indentation value corresponding to $F_{ad}$. Note that $F_{ad}$ is defined as positive in this equation.

**Differential rat lung ventilation**. Twelve pathogen-free male Sprague Dawley rats (350–450 g) mounted on the different experimental groups (6 for 2.1–0.1 Hz and 6 for 1.1–1.1 Hz). Animals were housed in controlled animal quarters under standard light, temperature, and humidity exposure. All experimental procedures were approved by the Ethical Committee for Animal Research of the University of Barcelona (Approval number 147/18).

Animals were anesthetized intraperitoneally using 20% urethane (10 mL/kg). After confirmation of deep anesthesia by tail and paw clamp, animals were tracheostomised and each lung of the rat was independently cannulated (16 G; BD Bioscience, San Jose, USA). After muscular relaxation with 0.4 mg/kg pancuronium bromide (Sigma Aldrich, St. Louis, MO) intravenously injected through the penile vein, each lung was connected to a customized small rodent ventilator, with a pressure sensor connected at the entrance of each canula to monitor animal ventilation according to the conditions explained below. Correct cannulation of each lung was confirmed by opening the chest wall and observing proper independent inflation and deflation of each lung. Animals were ventilated with a tidal volume of 21 mL/kg of animal weight (kg-bw) and with a positive end-expiratory pressure of 3 cmH2O. Control ventilation was set to a typical frequency of 1.1 Hz with the same tidal volume in each of the two lungs of the rat. To test the

effect of varying the ventilation frequency on YAP, a different ventilation frequency was applied to each lung while maintaining the control tidal volume. The left lung was ventilated at 0.1 Hz, and the right lung at 2.1 Hz. In this way, the animal received the same total minute ventilation, hence keeping $O_2$ and $CO_2$ blood gas levels thereby discarding any systemic effect induced by differential ventilation.

## Computational clutch model

*Summary of previous clutch model.* The computational clutch model was developed by adapting a previously described stochastic simulation, which we implemented to understand cell response to substrate rigidity[20]. Briefly, the model considers an actin filament that can bind to a given number of fibronectin ligands on the matrix $n_f$, through molecular links (clutches) composed of talin and integrin, with spring constant $k_c$. Fibronectin molecules are all connected in parallel to the substrate, represented by a spring with constant $k_{sub}$. The simulation begins with all clutches disengaged. At each time step, different events are allowed to occur stochastically, according to their respective rates. Unbound fibronectin molecules can bind integrins according to a loading rate $k_{on} = k_{ont} \cdot d_{int}$, where $k_{ont}$ is the true binding rate characterizing the interaction, and $d_{int}$ is the density of integrins on the membrane. Fibronectin molecules already bound to an integrin-talin clutch can experience two types of events. First, the integrin-fibronectin bond can unbind according to an unbinding rate $k_{off}$, which depends on force as a catch bond as experimentally measured[20,39]. Second, talin molecules can unfold according to an unfolding rate $k_{unf}$, which behaves as a slip bond also as experimental measured[40]. Because the load on each integrin may be shared between talin and other adaptor molecules, the force used to calculate unfolding is corrected by a factor $FR$, corresponding to the fraction of integrin-transmitted force experienced by talin.

At the end of each time step, the actin filament moves with respect to the substrate at a speed $v$, and total force on the substrate $F_{sub}$ is calculated imposing force balance:

$$F_{sub} = \frac{k_{sub} k_c \sum_{i=1}^{n_{bound}} x_i}{k_{sub} + n_{bound} k_c} \quad (4)$$

Where $x_i$ is the position of each bound fibronectin molecule, and $n_{bound}$ is the total number of bound molecules. Further, if a talin molecule has unfolded, mechanosensing is assumed to occur through adhesion growth, which is modeled by increasing integrin density $d_{int}$ by $d_{add}$ integrins/µm². If the integrin unbinds before talin unfolding, integrin density decreases by the same amount. However, integrin densities are only allowed to fluctuate between a minimum value $d_{min}$, and a maximum value $d_{max}$, reflecting the range between nascent adhesions and fully formed focal adhesions.

*Modifications to model stretch experiments.* Beyond this previously implemented model, we modified it in the following ways to specifically reproduce our stretch experiments:

1. First, we introduced a third event (apart from integrin unbinding and talin unfolding) that can occur in a given clutch. This event is the disruption of actin filaments, which had equivalent effects to integrin unbinding (that is, disengagement of the clutch, and reduction in integrin density). This event was only allowed to occur for clutches where talin unfolding (and thereby mechanosensing and reinforcement) had not occurred. We note that although this event is presented as the breaking of actin filaments, it may also include other events disrupting the actin cytoskeleton pulling on clutches, such as instance disruption of actin crosslinks. Thus, it is not straightforward to assign a given probability rate for the event as a function of force, and for simplicity, we simply modeled that this occurred instantaneously above a given threshold of force $F_{act}$. The value that best fit our data (142 pN) is in good agreement with experimentally reported values to break actin filaments[41].

2. Second, the speed $v$ of movement between actin and the substrate was not driven by actomyosin contractility (as done in our previous work), but by the applied triangular stretch signals: that is, a constant speed that changed direction twice for each period of the applied signal (with amplitude $A$ and frequency $f$). Thus, $|v| = 2Af$, and its sign changed every half cycle. As explained in the main text, the speeds derived from stretch were expected to be similar to actomyosin flows only in the mildest stretch condition (2.5% stretch, 0.125 Hz), so it is reasonable to assume that speeds were mostly driven by stretch across conditions.

3. Third, simulations were not carried out using a fixed time step as done previously, but by a Gillespie algorithm[79] which only executes one event per time step. That is, for each time step, we calculated stochastically the time at which each possible event for each clutch would occur, according to their respective rates. Then, the algorithm executes only the event occurring first. This was done to ensure that for the highest actin speeds applied (much higher than actomyosin-generated speeds considered in our previous work), potential very fast events were not missed by a fixed, too large time step.

4. Finally and for the same reason, we encountered the problem that for high actin speeds, calculation of rates for the different events was inaccurate. That is, if for instance a given clutch has just been formed and not yet been pulled by actin, its $k_{off}$ is that corresponding to zero force. However, if actin pulls

very quickly, by the time the algorithm predicts an unbinding event, force has risen significantly. This leads to a major change in $k_{off}$ within the same time step. To account for this, we corrected the different rates to take into account how they were affected by the loading rate. To this end, for each event we first calculated the probability density $p(F,L)$ as a function of applied force $F$, when force is ramped starting from zero at a loading rate $L$, as described previously[80]:

$$p(F, L) = \frac{k(F)}{L} \exp\left(-\int_0^F \frac{k(F)dF}{L}\right) \quad (5)$$

Where $k$ is the rate of the event in question ($k_{off}$ for integrin unbinding or $k_{unf}$ for talin unfolding). The units of $p$ are of force$^{-1}$, and it can be checked that $\int_0^\infty p(F)dF = 1$. Then, the mean force $F_m$ at which the event takes place can be calculated as:

$$F_m = \int_0^\infty F p(F)df \quad (6)$$

With $F_m$, one can readily calculate the lifetime of the event as $F_m/L$, and then define a corrected rate $k'$ as the inverse of the lifetime:

$$k' = \frac{L}{F_m}$$

As stated above, this expression applies to the particular case in which force loading starts at zero force, that is $k'$ corrects the value of $k(F=0)$ for a given applied loading rate $L$. In our simulations, at a given time step forces applied to a given clutch are not necessarily zero, and thus we need to consider the case of a force ramping from a given baseline force $F_b$. To do this, we simply imposed that $p(F < F_b, L) = 0$, and normalized the probabilities by $\int_{F_b}^\infty p(F)dF$ to maintain a total probability of 1.

At the end, instead of the initial rate $k(F)$, which depends only on force and assumes that force stays constant, we get a corrected rate $k'(F,L)$, which considers that force starts at value $F$ and then ramps up with a loading rate $L$. In all cases, we verified that for sufficiently low values of $L$, $k' = k$. This confirms the expected result that if the timescale of force loading is slower than that of event lifetime, rates are simply the ones you would obtain by assuming constant force.

Thus, for either $k_{off}$ or $k_{unf}$, calculating this corrected rate allowed to consider the effects of loading rate. For actin breaking events we note that this was not necessary, since we assumed the event to occur instantaneously above a threshold force.

*Model parameters.* All model parameters are described in Supplementary Table 4. Most parameters were taken exactly as in our previous work, where their choice and relationship to experimental values are discussed, along with a sensitivity analysis[20]. For $k_{sub}$ (substrate stiffness), we took values corresponding to the lower range employed in our previous work, which matches the 0.5 kPa polyacrylamide Young's modulus used here. Parameters specifically introduced or modified include:

– The frequency ($f$) and amplitude ($A$) of the stretch signal applied. Frequencies were chosen to reproduce the experimental range, and amplitude values provided good fits at values ranging from 1.5 to 3 µm. These values are of the order of the expected amplitudes of relative movements of cells ~20 µm in diameter being stretched by 2.5–20%. Interestingly, the range of amplitudes providing good fits in the model (1.5–3 µm) is narrower than the experimental range (2.5–20%), showing that the experimental sensitivity of cells is less than that predicted by the model.

– A parameter $d_{max}$ was introduced, setting the maximum value of integrin recruitment (and thereby adhesion growth). This was done to reproduce the experimental range in adhesion sizes. We note that in our previous model, the limit to adhesion growth emerged naturally from the stalling of actomyosin flows, once substrate forces reach the maximum forces that myosin motors can exert. In this model, flow speeds do not stall as they are imposed by stretch, and thus a limit to their growth has to be imposed.

– The number of integrins added each time mechanosensing takes place ($d_{add}$) was increased from 10 to 24 /µm² from our previous model. This was necessary to allow the model to respond within the timescale of fast stretch oscillations.

– Finally, as described above a parameter $F_{act}$ quantifying the force required to break actin filaments was introduced. This value falls within the range of reported experimental values[41].

Throughout simulations, all parameters except $A$ and $f$ were fixed, according to the values in Supplementary Table 4. To model the different experiments, only $A$ and $f$ were modified, according to the experimental conditions.

*Model output.* Model simulations were run for 1000 s, and then the average values of integrin densities $d_{int}$ were calculated. These values were then taken as a proxy of adhesion growth, and compared to adhesion length measurements in stretch

experiments. To compare the results, model values were scaled between the minimum and maximum experimental length values, and plotted in the same graphs in Fig. 2e. Note that this scaling was the same for all panels, and not modified to specifically fit each panel.

**Statistical analysis**. No statistical methods were used to determine sample size before the execution of the experiments. All independent datasets were first checked for normality using the d'Agostino-Pearson K2 normality test. Statistical tests used in each case are specified in figure legends, and were calculated using Graphpad PRISM software. All statistic tests were two-sided, and included adjustments for multiple comparisons. All central tendency values are mean and the error bars shown are standard error of the mean. Significance is considered for $p < 0.05$. In all cases, at least two independent experimental repeats were carried out for each condition. The exact number of samples ($n$) is specified in figures or in Supplementary Table 3.

**Reporting summary**. Further information on research design is available in the Nature Research Reporting Summary linked to this article.

## Data availability
The authors declare that all data supporting the findings of this study are available within the paper and its Supplementary Information files. Source data are provided with this paper.

## Code availability
Custom scripts used in the manuscript are uploaded as supplementary material (Supplementary Software 1).

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

## Acknowledgements

This work was supported by the Spanish Ministry of Science and Innovation (PID2019-110298GB-I00, PGC2018-099645-B-I00), the European Commission (H2020-FET-PROACT-01-2016-731957, and the Marie Sklodowska-Curie Grant Agreement No. 798504 to A.E.-A.), the Generalitat de Catalunya (2017-SGR-1602), Fundació la Marató de TV3 (201936-30-31 and 201903-30-31-32), the European Research Council (ERC-Adv 883739 to X.T.), the prize "ICREA Academia" for excellence in research to P.R.-C., and "la Caixa" Foundation (Agreement LCF/PR/HR20/52400004). S.H and T.B. were supported by the German Science Foundation (EXC 1003 CiM, Cells in Motion), the European Research Council (Consolidator Grants 771201, PolarizeMe), and the Human Frontier Science Program (HFSP grant RGP0018/2017). A.E.M.B. was supported by a Sir Henry Wellcome Fellowship (210887/Z/18/Z). IBEC is the recipient of a Severo Ochoa Award of Excellence from the Spanish Ministry of Science and Innovation. We thank V. Gonzàlez-Tarragó for assistance in experiments, set-up implementation, and discussions. We thank the members of P. Roca-Cusachs, X. Trepat, T. Betz, I. Almendros, and R. Farré laboratories for technical assistance and discussions. We thank M. Brandt and D. Navajas for their technical assistance and discussions. We thank A. Lahiguera and C. Ureña for discussions.

## Author contributions

P.R.-C. conceived the study; I.An., B.F., S.H., A.-L.L.R., R.F., T.B., I.Al., and P.R.-C. designed the experiments; I.An., B.F., X.Q., and Z.K. performed the experiments; I.An., B.F., N.C., X.T., Z.K., A.E.M.B., A.E.-A., T.B., and P.R.-C. analyzed the experiments; and P.R.-C. performed the modeling work and wrote the manuscript. All authors commented on the manuscript and contributed to it.

## Competing interests

The authors declare no competing interests.
