## [Peer Review File · Nature Communications]

REVIEWER COMMENTS

Reviewer #1 (Remarks to the Author):

This manuscript examines the dependence of cellular mechanics on force loading rate. A key strength of the study is the creative combination of several experimental paradigms ranging from cultured cells to whole organs. Weaknesses include unclear biological significance, unexplained differences in results across the various systems, imprecision in language that hinders understanding, and somewhat limited mechanistic depth. One would also like to see a clearer expression of how this work fits into the 20+ year literature on the frequency-dependent rheology of cells. The work is commendable for its rigor and use of multiple experimental systems, but at the end of the study I have to admit that I have a difficult time pinning down what new information I've learned or why it's important.

1. The statement that the loading rate is the product of the deformation speed and stiffness is of course true for passive stretch of a linear spring but needs additional explanation when applied to a cell actively deforming a two-dimensional material. Are there TFM or micropillar studies the authors can cite empirically demonstrating that loading rate (force/time) does indeed increase with matrix stiffness (force/area)?
2. In Fig 1c the differences in YAP localization in the images aren't as dramatic as the plot would suggest. For example, YAP appears quite nuclear even in the NS image.
3. The discrepancy between the substrate stretch and optical tweezer-based frequency sweeps are very interesting and could be discussed more – why is the frequency response biphasic when stretch is applied to the whole cell and monotonic when stretch is applied locally to a few adhesions? Are the same governing mechanisms at play across the various systems? The same question comes up when interpreting the AFM and lung studies.
4. The AFM results are very interesting, but the interpretation of the data is very confusing. Typically stiffness values are determined by fitting indentation curves to a model of contact mechanics (e.g. Hertz). Here stiffness is apparently inferred from the retraction curve, which is both nonstandard and made much more complex because of the need to break adhesions. For example, the force curve in 1e shows a region of constant force between 1-2 microns, which is often associated with extension of a membrane tether (e.g. Sun+ BJ 89:4320, 2005). Which portions of the curve are being used to determine stiffness, and are the authors sure that the measurement reflects fluidization of the entire cell versus fluid-like behavior in whatever microscale portion of the cell is being probed in the AFM measurement?
5. What does “fluidization” mean from a molecular point of view, and how are those molecular events related to the frequencies used in the experiments?
6. Throughout the paper, “mechanosensing” is often implied to be a binary phenomenon (either happening or not), with YAP nuclear localization serving as the key readout. This is problematic on several levels, because the YAP changes seem more graded than binary (see earlier comment on Fig 1), and there are factors beyond force application that regulate YAP trafficking. The language should

be revised to be more precise.

7. In general, one is left with little sense of how to judge the YAP N/C ratios, particularly since the dynamic range tends to vary across paradigms (e.g. 1.2-2 for stretch, 0.8 – 1.5 for the lung studies). Are the changes observed expected to be biologically important in terms of YAP co-transcriptional activity? Is there some threshold above which one should expect significant phenotypic changes?

8. It would be helpful to have a bit more mechanistic insight behind the observations, particularly for the biphasic frequency response to stretch and the fluidization observed in AFM. Actin polymerization, as hypothesized in the discussion to govern the responses, seems like a somewhat broad and obvious candidate.

9. There have been a number of excellent MTC studies on how frequency/loading rate affect viscoelastic properties, including work from the authors' local colleagues (Puig-de-Morales, *J Appl Physiol* 91:1152, 2001) and early MTC papers from Jeff Fredberg, Don Ingber and Ning Wang. The authors should discuss how their results square with these earlier studies, with emphasis on the new conceptual contributions of the current study.

Reviewer #2 (Remarks to the Author):

Summary. The authors show that loading rate, or the rate that force is applied to cells, is a driving force for mechanosensing. By modulating loading rate through cell stretching experiments, the authors observe that loading rate correlates with YAP and paxillin mechanosensing responses in a talin-dependent manner. Low levels of stretch, and therefore loading rate, fail to induce YAP localization to the nucleus and focal adhesion growth. Above a loading rate threshold, this mechanosensitive signaling is lost, presumably through actin cytoskeleton fluidization. The authors use an optical tweezer method to eliminate the possibility that this effect is due to changes in cell deformation rather than force loading. This experiment revealed that mechanosensing is preserved at the local level, while fluidization only occurs at the whole cell level. Additionally, the authors use an in vivo mechanical ventilation model to confirm their findings on the whole-organ level.

Main Criticisms

1. Figure 3: The mechanisms by which fluidization occurs remain unclear. To address this shortcoming, the authors should do the following: Create a clear definition of fluidization. To make conclusions in the fluidization experiments (Figure 3), the authors should identify a way to quantify actin organization. For instance, they can track displacement of microbeads adhered to the cytoskeleton over the course of the experiment. In addition, it would be helpful if authors added similar images in the supplemental figures so that we can see that the images chosen are representative. There is insufficient information regarding the organization of the cytoskeleton network. Blebbistatin is not an ideal agent for determining whether actin disruption is sufficient for fluidization. Blebbistatin targets myosin II and is therefore not specific enough for use in this context. Cofilin, a protein required for the fluidization response, can be knocked down to complement the experiments done with blebbistatin, and to ensure that force loading is leading to a fluidization event.

2. The vimentin filaments are linked to paxillin-rich focal adhesion contacts, a region also rich in the

motor protein myosin II. What is the role of intermediate filament network on loading rate, particularly in the fibroblast in vitro model? In the lung, the environment would be subjected to repeating loadings. It was recently shown (<https://doi.org/10.1073/pnas.1903890116>), that the vimentin plays an essential role in maintaining the resilience of the cytoplasm because of its high yielding strain, while the rest of cytoplasmic components are greatly softened or even disassembled. If this is in fact what is happening how does an intact vimentin network effect the translocation of YAP and paxillin?

3. Figure 4: The in vivo approach shown alters frequency of ventilation and examines the impact on YAP translocation. High frequency ventilation (HFV) is a ventilatory strategy that utilizes a form of mechanical ventilation that typically combines very high respiratory rates (>60 breaths per minute) with tidal volumes that are smaller than the volume of anatomic dead space. The clinical rationale for this type of ventilation is that gas exchange is optimized by utilizing small tidal volumes with minimal alveolar stretch. To test the effect of varying the ventilation frequency on YAP, the authors applied a different ventilation frequencies and claimed that tidal volume was equal in each lung. The author will need to provide pressure-volume curves to support this claim. The left lung was ventilated at 0.1 Hz, and the right lung at 2.1 Hz. According to the authors, the animal received the same total minute ventilation, hence keeping O₂ and CO₂ blood gas levels thereby discarding any systemic effect induced by differential ventilation. It would be quite surprising that the right and left lungs had equal pO₂ and pCO₂ levels. The authors need to provide data demonstrating this important factor, particularly as YAP translocation to the nucleus has been shown to be oxygen dependent. Additionally, the authors should consider measuring alveolar deformation with the given ventilation volumes and frequencies. Ensuring that each alveolar deformation is similar in magnitudes would allow the authors to link (or not) that YAP nuclear localization is due to stretch alone. The authors state the in Figure 4, the cells in the alveoli are mostly epithelial and endothelial cells. In contrast, the prior figures use fibroblast; different cell types may respond differently to loading rate. It would be helpful if the authors co-immunostained and determined which cells in the lung had YAP translocate to the nucleus. Additionally, it would be helpful to have link between the in vitro and in vivo experiments (e.g. same cell type). This is important because different rates of cyclic stretch can cause cell proliferation and death (Liu et al., *AJP*, 1992; Tschumperlin et al., *AJRCCM*, 2000). Given this information, it would be useful to know the specific functional outcomes (proliferation, apoptosis, collagen production) of force loading rate on cells both in vitro and in vivo.

Minor comments:

The authors need to report ANOVA test statistics; it is unclear what is significant by looking at the figures and figure captions.

Not formatted in journal style (just "Main"; not "Introduction," "Results," "Discussion," etc.)

Reviewer #3 (Remarks to the Author):

In this manuscript, González-Tarragó et al. investigate how the force loading rate impacts cellular mechanotransduction. This is an important yet underexplored aspect of cell biology, and thus the current study is very timely. Using a number of different complementary experimental approaches, the authors convincingly demonstrate a force rate dependency of mechanotransduction and

connect it to fluidization/damage of force-transmitting proteins in cells (see below). The experiments are elegant, the paper is well written, and most of the conclusions are justified. The data reveal a new mechanotransduction mechanism that will be highly relevant for many future studies in different fields.

There are two major points, however, which the authors should address.

1) I am not sure if what the authors observed really corresponds to a fluidization of cells. If I understand it correctly, all observations can simply be explained by structural damage (or plastic deformation after the yield point), which is, technically speaking, not the same as fluidization. Fluidization might be a consequence of structural damage, but it is not necessarily the cause of the observed behavior. It's just a terminology issue, but an important one. As fluidization was not shown directly anyway (e.g., a change in the cells' viscosity), I'd strongly recommend to re-word the manuscript and avoid using the term 'fluidization'.

2) I have some questions about the analysis of the AFM experiments. The authors currently simply fit a linear function through 2 fixed points on a highly non-linear curve, concluding that cells liquify (or better: likely undergo structural damage) at a loading rate between 5 and 6 $\mu\text{m/s}$. Looking at the raw data in Fig. 3b, however, I would argue that large parts of the slopes in these two plots (the more reliable parts further way from the point of detachment) look nearly identical, and that the fits are actually rather poor. How would the stiffness distributions look like if only data above 0.5 nN (or a similar threshold) would be analysed? And how if a nonlinear model, such as the Hertz model, would be used to analyse the data? Would Fig 3b still suggest a similar tendency? If not, the authors should re-think their conclusions.

Furthermore, the authors should provide information about what the data points show (mean \pm SEM?) and about the statistical analysis in the figure caption. Assuming that they used a multiple comparisons test in Fig. 3c as mentioned, amongst other tests, in the methods, is the difference between 5 and 6 $\mu\text{m/s}$ the only one that is statistically significant? So is there no difference between 2 and 4 and 8 $\mu\text{m/s}$, for example? What does that mean?

Minor points:

Abstract: I would omit 'In contrast' in sentence 4, as it could well be in addition.

Main text: 'the fundamental mechanical variables that cells sense and respond to are unknown'. I would disagree with this statement and suggest to change it to ... not fully understood.

Page 4 paragraph 1: 'applying a very mild stretch (2.5% biaxial stretch, applied cyclically with a triangular 0.125 Hz wave for 1 h) leads to a deformation speed of ~ 60 nm/s, of the same order of magnitude than internally generated actomyosin flows.' It is confusing to compare these two time scales here, as they are not related (or, if they are, then rather inversely): An increase in deformation speed leads to more mechanotransduction, while an increase in actin flow rates generally leads to less mechanotransduction. I'd suggest to omit the second part of this sentence.

Page 6 paragraph 1: 'Thus, at high levels of stretch or stretch rates, cytoskeletal softening would reduce the loading rates being applied to adhesions, since loading rates depend not only on the rate of deformation but also on the stiffness of the structure being deformed.' I agree with this statement, but wouldn't this also lead to a change in load amplitude, which might be partly contributing to what the authors see as well? The authors might briefly want to discuss this.

It would be nice to add a paragraph to the discussion about how the loading rate may impact structural damage within cells.

Statistical analyses should also be conducted for supplementary figures. If they were done and no differences observed, this should be stated as well.

Reviewer #4 (Remarks to the Author):

The authors indicate that the loading rate of force application is a key driver of mechanosensing. However, above certain thresholds the cytoskeleton collapses/fluidizes/softens, and this prevents strong adhesion to the ECM, as focal adhesions and the cytoskeleton lack reinforcement. In isolated rat lungs *ex vivo*, they demonstrate the relevance of the loading rate (but not of fluidization). The key message here is that what cells read are force dynamics, rather than absolute values or thresholds.

I have few, mainly conceptual questions, some requiring additional experiments, and other points just discussion:

- 1) They assume (line 74) that cell-substrate attachment and force transmission occurs largely at the cell periphery. Is it so? A recent paper by Vogel and colleagues (Shiu et al 2018) has instead shown that the central (perinuclear) area of the cell displays the more robust traction force, and is mainly responsible for YAP nuclear accumulation. How is this changing their conclusions?
- 2) They show the relevance of the loading rate, either by increasing ECM stiffness, or by increasing the frequency of deformations using stretching pulses on cells with a stretching device. For example they show that the same effect on YAP or Focal adhesion (FA) can be obtained either by 4x stiffness (E) or by keeping E constant (of a soft ECM) and increasing the frequency from 0.125 to 1 Hz. This is certainly interesting and intriguing. But how does it work? in the revised molecular clutch model (Elosegui-Artola 2016) the key determinant of mechanosensitivity is whether the resistive force of the ECM can be transmitted to talin before the integrin-ECM bond dissociates. In that model, on a soft ECM, integrins dissociate from their ECM ligands faster than (and thus before) any engagement of actin to integrin (through clutching molecules). Here: How is stretching rate and frequency changing that? Are these stabilizing/increasing the binding affinity/avidity of integrin for ECM? or are these inputs making molecules such as talin or vinculin loading at faster frequency (compared to the control "still" soft ECM) on integrin, leading to reinforcement? The authors should offer some explanations or interpretations.
- 3) Is this connected to cell shape, such that a cell that is challenged to spread by stretching pulses compatible with the lifetime of integrin-ECM bonds responds by increasing its own pulling, with faster clutch loading, with talin deformation and vinculin-mediated reinforcement, as it occurs in cells on stiffer ECM? This is not shown, but could be interesting to demonstrate on cells experiencing the stretching device, and not only using beads in Fig2.
- 4) The simplest interpretation is that mechanosensing occurs only above (absolute?) thresholds that are or not reached depending on for how long/how often cells experience the maximal amount of (tolerated) stretch in a given time window. Square stretches gain their target before and more robustly than triangular ones, and increasing E further facilitates the goal. As if cells are able to integrate all these inputs and count them, accumulating them. Please comment.
- 5) they also report that at very high stretch rates/ stretch magnitudes cells lose the connection with

their ECM, as this causes softening of their cytoskeleton. My concern here is why is this not happening on cells that are "normally" attached to glass or plastic? Is this connected to the need of some minimal and required timing in the dynamic of FA function - globally, at the cell level - that once overruled by experimental stretching leads to failure of all clutches and loss of traction?

6) At the end, to what extent is this cell relaxation phenomenon really physiological? It is potentially intriguing, yet the lung ventilation experiments do not show any fluidization, for example, but only increased response to increased stretching frequency. What about other cells and tissues with much faster stretching and relaxation rates, such as heart or vessels?

7) They point at actin filaments to explain fluidization. But what about microtubules or intermediate filaments?

Response to reviewer comments

Here we include a point-by-point reply to all reviewer comments, comments are in *grey*, and our replies in black.

Reviewer #1 (Remarks to the Author):

This manuscript examines the dependence of cellular mechanics on force loading rate. A key strength of the study is the creative combination of several experimental paradigms ranging from cultured cells to whole organs. Weaknesses include unclear biological significance, unexplained differences in results across the various systems, imprecision in language that hinders understanding, and somewhat limited mechanistic depth. One would also like to see a clearer expression of how this work fits into the 20+ year literature on the frequency-dependent rheology of cells. The work is commendable for its rigor and use of multiple experimental systems, but at the end of the study I have to admit that I have a difficult time pinning down what new information I've learned or why it's important.

As we detail in response to the specific comments below, we think we have resolved all the concerns mentioned. Regarding the fundamental question of what is new or important in our manuscript, we believe that the fundamental message of our work is that the force loading rate (and not simply the frequency of an oscillatory signal) drives mechanosensing (and not simply rheology). Importantly, by developing a computational clutch model in this revision, we now also show that this response can be explained naturally from the force-dependent properties of key mechanosensing elements, i.e., integrins, talin, and the actin cytoskeleton. This demonstrates that the same conceptual framework can explain cell mechanosensing in response to externally applied forces or to passive mechanical parameters such as substrate rigidity, which we think is a major advance in the field.

1. The statement that the loading rate is the product of the deformation speed and stiffness is of course true for passive stretch of a linear spring but needs additional explanation when applied to a cell actively deforming a two-dimensional material. Are there TFM or micropillar studies the authors can cite empirically demonstrating that loading rate (force/time) does indeed increase with matrix stiffness (force/area)?

This is a very interesting question. To our knowledge, there are no studies analyzing this systematically. However, one can extract indirect data for instance from Ghassemi et al., PNAS 2012. In this paper, they measure the dynamics of force generation in cells pulling on pillars of different sizes and stiffness (in fact, the same cells used in our paper). By considering the reported values of pillar displacements, stiffness, and time scales of deformation (Fig. 4 of the paper), one can estimate the range of loading rates. Indeed, as pillar stiffness increases by one order of magnitude from the softest to the stiffest, the loading rate also increases by an order of magnitude. However, we would like to point out that using traction forces on substrates with different stiffness to demonstrate a role of loading rate is very complicated: if one considers an ideal scenario in which a cell forms a nascent adhesion to the substrate and starts pulling at a given speed (similar in fact to the Ghassemi et al. conditions), it is clear that the loading rate will be higher for stiffer substrates. This is in fact at the foundation of the molecular clutch model, and it is the principle that explains why the model predicts different responses for different stiffness. However, after this initial event loading rates will be affected in non trivial ways by many factors, including for instance increased cell-ECM adhesion due to focal adhesion formation, distribution of forces among different adhesions, re-structuring of

the cytoskeleton, or reaching the maximum contractile forces that a cell can exert. Molecular clutch models take some of these factors into account to predict resulting effects on forces and adhesions, but due to the coupling between the different factors, the role of loading rate can not be determined univocally. This was precisely the aim of this work, to apply external forces in a controlled way to directly evaluate the role of the loading rate.

To clarify this, we now cite the Ghassemi paper in the introduction (page 3). Additionally and as explained below, we have developed a new implementation of the clutch model to demonstrate that the same principles can be employed to understand passive sensing of ECM, and externally applied forces.

2. In Fig 1c the differences in YAP localization in the images aren't as dramatic as the plot would suggest. For example, YAP appears quite nuclear even in the NS image.

After checking the figure, we have realized that images could give ambiguous impressions of YAP localization, simply because nuclear locations were not marked. In the example of the NS image that the reviewer mentions, the brighter region of the cell that could be interpreted as a YAP-rich nucleus does not in fact correspond to the nucleus. To correct this, we have now overlaid the outline of the nucleus (as obtained in corresponding Hoechst stainings) on the images.

3. The discrepancy between the substrate stretch and optical tweezer-based frequency sweeps are very interesting and could be discussed more – why is the frequency response biphasic when stretch is applied to the whole cell and monotonic when stretch is applied locally to a few adhesions? Are the same governing mechanisms at play across the various systems? The same question comes up when interpreting the AFM and lung studies.

To clarify the mechanism we propose to interpret our results, we have now developed a computational clutch model. Based on our previous work (see Elosegui-Artola et al., Nat. Cell Biol. 2016), this model considers progressive force application to links between actin, talin, integrins, and fibronectin i.e., “clutches”). Then, it considers how talin unfolding and integrin-fibronectin unbinding depend on force (based on experimental single molecule data). As force builds in each timestep, the montecarlo simulation evaluates whether talin will unfold, or integrins will unbind. If talin unfolds before the integrin unbinds, we assume that there is a mechanosensing event, which leads to integrin recruitment (i.e., adhesion growth in experiments). As a modification from our previous model, here we introduce that i) force on clutches does not arise from actomyosin contractility, but from an externally imposed stretch, and ii) the clutch can be disengaged not only by integrin unbinding, but also by actin cytoskeleton disruption (i.e., fluidization) above a threshold force. Of note and as discussed further in response to point 5 below, this event in the model cannot distinguish between different potential events, such as breaking of actin filaments, or severing of actin crosslinks, for instance. However, the model provided a good fit to the data by assuming a force of about 140 pN, in reasonable agreement with reported experimental values for the breaking of actin filaments (Kishino and Yanagida, Nature 334, 74–76, 1988). Importantly, the model does not assume any dependence on loading rate per se. Simply, if loading rates are low, forces stay long enough in a low regime where integrin unbinding is more likely than talin unfolding. If loading rates increase, forces quickly reach a regime in which talin unfolding is more likely, triggering mechanosensing. For very high loading rates, the force required for fluidization is reached

before talin unfolding can occur. In the case of optical tweezers experiments, fluidization is prevented for two reasons. First, tweezers are limited to forces below 100 pN. Second, measurements are carried out in the lamellar region, with an actin network much more structured than on the rounded cell phenotype found on soft substrates before stretch. Thus, forces applied to beads are likely distributed among many filaments, reducing the likelihood of fluidization. Consistently with this, loading-rate induced softening in AFM experiments (indicative of fluidization) is observed when rounded cells are pulled, but not when beads attached to lamellipodia are pulled. In the case of lungs, and although it is hard to assess in detail due to the much less controlled nature of the setup, it is tempting to speculate that tissue architecture has built-in mechanisms to distribute loads and prevent disruption of the actin cytoskeleton.

We now present the model throughout the text (see detailed description in methods), and its predictions are shown in figure 2.

4. The AFM results are very interesting, but the interpretation of the data is very confusing. Typically stiffness values are determined by fitting indentation curves to a model of contact mechanics (e.g. Hertz). Here stiffness is apparently inferred from the retraction curve, which is both nonstandard and made much more complex because of the need to break adhesions. For example, the force curve in 1e shows a region of constant force between 1-2 microns, which is often associated with extension of a membrane tether (e.g. Sun+ BJ 89:4320, 2005). Which portions of the curve are being used to determine stiffness, and are the authors sure that the measurement reflects fluidization of the entire cell versus fluid-like behavior in whatever microscale portion of the cell is being probed in the AFM measurement?

We apologize since this experiment was not clearly explained in the manuscript. The aim of this experiment was to assess whether stretching cells led to their softening (i.e., fluidization). To this end, the portion of the curve analyzed had to be the retraction curve, in which the cell is being stretched. More specifically, we fitted the retraction curve from the moment in which retraction starts, until the moment in which the maximum pulling force is achieved. After that maximum pulling force, the cell starts indeed breaking adhesions and detaching, and force starts decreasing until it reaches zero. As the reviewer points out, this makes the interpretation much more complex, and events like the region of constant force between 1-2 μm are observed. Thus, this region of the curves was not included, and we are confident that our analysis reflects the mechanical properties of the cell as a whole and not of a pulled tether, for instance.

However, to further improve our analysis and as also suggested by other reviewers, we have now improved our fits in two ways. First, instead of a linear fit, we have used a DMT model, which is a modification of the Hertz contact model to take adhesion into account. Essentially, the curve is fitted to a Hertz contact model, but with the modification that the curve is offset by the maximum pulling force (force of adhesion). Second, we have restricted the fit to the region of the pulling curve in which the cell is under stretch, and not compression (i.e., negative force values in figs. 4b and 4e). That is and taking the curves in fig. 4b as an example, we fitted the curve starting from zero force and ending in the maximum pulling force. We have now clarified this in the figure itself, in the text, and in the methods.

5. What does "fluidization" mean from a molecular point of view, and how are those molecular events related to the frequencies used in the experiments?

We apologize for not sufficiently explaining this in our manuscript. “Fluidization” is a term commonly used to describe the well-known phenomenon by which stretching cells with a sufficiently high amplitude (or sufficiently high frequency) leads to their softening. This behaviour can’t be pinpointed to a specific molecular event, but rather responds to an overall disruption of the cytoskeleton which likely involves several different molecular interactions (see Trepal et al., Nature 2007, doi.org/10.1038/nature05824). Accordingly, and as shown by Trepal et al. and others, fluidization is modulated by a wide range of molecular perturbations targeting different cytoskeletal elements. We would like to clarify that in its original definition (for instance in Trepal et al., and Kollmannsberger and Fabry, doi.org/10.1146/annurev-matsci-062910-100351) fluidization was meant to describe both cytoskeletal softening and a more viscous-like behaviour, hence its name. However, it has later been used widely to refer simply to cytoskeletal softening upon stretch (see for instance Harris et al., doi.org/10.1073/pnas.1213301109, or Nava et al., doi.org/10.1016/j.cell.2020.03.052). In our case, we are also simply referring to cytoskeletal softening. Although we acknowledge the imprecision, we have preferred to keep this term in the manuscript to best link it to related literature. However, we have clarified this potential confusion in the text (page 5).

To clarify how fluidization fits into our experiments, we have developed the molecular clutch model described in response to point 3 above. That is, “fluidization” in this context means actin cytoskeleton disruption, which we model as a disengagement of the clutch above a force of 140 pN. As stated above and although fluidization can’t be pinpointed to a specific molecular event, this force is in reasonable agreement with experimental data for the breakage of actin filaments, and further supported by the fact that our fluidization response is eliminated by jasplakinolide, which stabilizes actin filaments. In terms of frequencies, higher frequencies increase loading rates, and increased loading rates mean that bonds (clutches) will reach higher forces before detaching. This is simply because a given level of force will be reached faster if the loading rate is high, allowing less time for detachment to occur. As explained above, progressively increasing loading rates will lead to regimes where the first event to occur is integrin unbinding (low rates), talin unfolding (medium rates), and fluidization (high rates).

6. Throughout the paper, “mechanosensing” is often implied to be a binary phenomenon (either happening or not), with YAP nuclear localization serving as the key readout. This is problematic on several levels, because the YAP changes seem more graded than binary (see earlier comment on Fig 1), and there are factors beyond force application that regulate YAP trafficking. The language should be revised to be more precise.

We fully agree with the reviewer in that mechanosensing is a graded phenomenon, which is reflected indeed in the progressive changes of both YAP localization and adhesion length. This is reflected both in experiments, and also now in the theoretical predictions of our model. We also note that except in the in vivo experiments (where adhesion length measurements were not possible), throughout the paper we use both YAP and adhesion quantifications to assess the degree of mechanosensing. Both are used as independent indicators of mechanosensing, and both show the same trends. To clarify this issue, we have now corrected the text to refer to progressive changes in both YAP localization and adhesion size.

7. In general, one is left with little sense of how to judge the YAP N/C ratios, particularly since the dynamic range tends to vary across paradigms (e.g. 1.2-2 for stretch, 0.8 – 1.5 for the lung studies). Are the changes observed expected to be biologically important in terms of YAP co-

transcriptional activity? Is there some threshold above which one should expect significant phenotypic changes?

To address this issue, we have now carried out measurements of proliferation rates as a function of stretch (see supplementary fig. 1c). Indeed, proliferation is a known major downstream effect not only of YAP transcriptional activity, but of integrin-mediated mechanosensing in general. Our results show that proliferation follows the same trends observed both for adhesion lengths and YAP, i.e. a biphasic response. Unfortunately, this could not be checked at the level of lungs, since times of stimulation are limited to about one hour due to lung damage induced by overventilation at longer time scales, and this time was insufficient to observe a transcriptional effect.

8. It would be helpful to have a bit more mechanistic insight behind the observations, particularly for the biphasic frequency response to stretch and the fluidization observed in AFM. Actin polymerization, as hypothesized in the discussion to govern the responses, seems like a somewhat broad and obvious candidate.

We believe that our newly developed clutch model now clarifies this, as addressed in response to the previous comments of the reviewer. Regarding the fluidization response, we want to clarify that our contribution is to demonstrate that it occurs in the context of our experiments, and not to fully elucidate the complex mechanisms behind the phenomenon. As discussed above, there is a broad body of literature examining how cells and cytoskeletal networks soften/fluidize in response to stretch, and the many biophysical and molecular factors involved.

9. There have been a number of excellent MTC studies on how frequency/loading rate affect viscoelastic properties, including work from the authors' local colleagues (Puig-de-Morales, J Appl Physiol 91:1152, 2001) and early MTC papers from Jeff Fredberg, Don Ingber and Ning Wang. The authors should discuss how their results square with these earlier studies, with emphasis on the new conceptual contributions of the current study.

We thank the reviewer for bringing out these relevant publications. These papers showed the important finding that cells exhibit higher resistance to force (typically assessed through the complex shear modulus G^*) when deformed at higher frequencies. The main conceptual difference between these studies and our work is that the main focus of our work is not cell rheology, but mechanosensing, i.e. the transduction of force into downstream responses (in our case, focal adhesion formation and YAP nuclear translocation). In this regard, an important aspect of our setup is that we apply stretch to cells seeded on soft substrates, so that no mechanical responses are triggered prior to mechanical stimulation. Mechanosensing is not addressed in these early MTC papers, and thus the respective contributions of those papers and our work are very different. Further, we combine different techniques and protocols (i.e., triangular versus square signals) to dissect a specific role of the loading rate (and not just frequency), which was not done in these earlier works either.

That being said, the frequency dependence of cell rheology measured with MTC can certainly be compared to our AFM results, which show that increasing deformation speeds first increase, and then decrease, apparent cell stiffness. The initial phase of stiffness increase (with either deformation speed or frequency) is fully consistent with MTC results, while the second phase of decrease was not observed in those studies. Most likely, this is because the range of

forces applied by MTC is of a few pN, too low to trigger fluidization. This is consistent with our optical tweezers results, where the force range is similar and fluidization was not observed. We now address these aspects in page 8 of the manuscript.

Reviewer #2 (Remarks to the Author):

Summary. The authors show that loading rate, or the rate that force is applied to cells, is a driving force for mechanosensing. By modulating loading rate through cell stretching experiments, the authors observe that loading rate correlates with YAP and paxillin mechanosensing responses in a talin-dependent manner. Low levels of stretch, and therefore loading rate, fail to induce YAP localization to the nucleus and focal adhesion growth. Above a loading rate threshold, this mechanosensitive signaling is lost, presumably through actin cytoskeleton fluidization. The authors use an optical tweezer method to eliminate the possibility that this effect is due to changes in cell deformation rather than force loading. This experiment revealed that mechanosensing is preserved at the local level, while fluidization only occurs at the whole cell level. Additionally, the authors use an in vivo mechanical ventilation model to confirm their findings on the whole-organ level.

Main Criticisms

1. Figure 3: The mechanisms by which fluidization occurs remain unclear. To address this shortcoming, the authors should do the following: Create a clear definition of fluidization.

We apologize for not sufficiently explaining this in our manuscript. “Fluidization” is a term commonly used to describe the well-known phenomenon by which stretching cells with a sufficiently high amplitude (or sufficiently high frequency) leads to their softening. This behaviour can’t be pinpointed to a specific molecular event, but rather responds to an overall disruption of the cytoskeleton which likely involves several different molecular interactions (see Trepats et al., *Nature* 2007, doi.org/10.1038/nature05824). Accordingly, and as shown by Trepats et al. and others, fluidization is modulated by a wide range of molecular perturbations targeting different cytoskeletal elements. We would like to clarify that in its original definition (for instance in Trepats et al., and Kollmannsberger and Fabry, doi.org/10.1146/annurev-matsci-062910-100351) fluidization was meant to describe both cytoskeletal softening and a more viscous-like behaviour, hence its name. However, it has later been used widely to refer simply to cytoskeletal softening upon stretch (see for instance Harris et al., doi.org/10.1073/pnas.1213301109, or Nava et al., doi.org/10.1016/j.cell.2020.03.052). In our case, we are also simply referring to cytoskeletal softening. Although we acknowledge the imprecision, we have preferred to keep this term in the manuscript to best link it to related literature. However and as requested, we have clarified this potential confusion and our definition of the term in the text (page 5).

In our work, our aim was not to elucidate the complex biophysical and molecular factors behind fluidization, which have already been analysed in the papers mentioned above, and several others. Instead, our aim was to show that fluidization does occur in our context. To further clarify how this fits into our data, we have developed a computational clutch model to account for the effect of changes in frequency and amplitude of stretch, integrin unbinding, talin unfolding, and fluidization. The effects of integrin unbinding and talin unfolding are taken from experimental data, and introduced in the exact same way as in previous modelling by our group (Elosegui-Artola et al., *Nat. Cell Biol.* 2016). The effect of fluidization is modelled as a disengagement of the “clutch” (i.e., the link between the substrate, integrins, and actin) above a threshold force. Interestingly and although fluidization can’t be pinpointed to a specific molecular event, we obtained good fits with a threshold force of about 140 pN, in reasonable agreement with experimental data for the breakage of actin filaments (Kishino and Yanagida, *Nature* 334, 74–76, 1988).

The model and its interpretation are now introduced throughout the text, and its results are shown in figure 2.

To make conclusions in the fluidization experiments (Figure 3), the authors should identify a way to quantify actin organization. For instance, they can track displacement of microbeads adhered to the cytoskeleton over the course of the experiment. In addition, it would be helpful if authors added similar images in the supplemental figures so that we can see that the images chosen are representative.

To address this point, we have quantified actin organization by computing its degree of anisotropy, by using a previously described image analysis tool (Boudaoud et al., Nat. Protoc. 9, 457–463, 2014). As actin progressively goes from a disordered organization to an ordered one with stress fibers, this reflects in an alignment of actin structures (fibers), which translates in a higher anisotropy of the image. As expected and following the trends of both adhesions and YAP, actin anisotropy peaks at a frequency of 1 Hz (triangular signal), and decreases when a square signal is applied. The new quantifications and statistical analyses are shown in fig. 2b. Since now we have quantified the phenotype, we believe it is not necessary to add further examples of the images in the supplement, but if the reviewer still deems it important we will be happy to do so.

There is insufficient information regarding the organization of the cytoskeleton network. Blebbistatin is not an ideal agent for determining whether actin disruption is sufficient for fluidization. Blebbistatin targets myosin II and is therefore not specific enough for use in this context. Cofilin, a protein required for the fluidization response, can be knocked down to complement the experiments done with blebbistatin, and to ensure that force loading is leading to a fluidization event.

We would like to clarify that our use of blebbistatin was not to test fluidization, but rather to understand the differences between stretch and optical tweezers experiments. Indeed, blebbistatin disrupts focal adhesions and thereby long-range force transmission across the cell body. As such, this should disrupt force buildup in response to cell stretch. However, blebbistatin does not eliminate the actin meshwork in cell lamellipodia, where optical tweezers measurements were taken. Thus, the prediction was that blebbistatin should abrogate the effect observed in stretch but not optical tweezers experiments, and this was indeed the case. To test the specific role of fluidization, we used another drug, jasplakinolide, which does specifically stabilize actin filaments. As expected, treating cells with jasplakinolide rescued cell response to stretch at high loading rates. Indeed, exposure to a 1 Hz square signal, which had no effect under control conditions, led to the focal adhesion growth for cells treated with jasplakinolide (Fig. 2c). We have now further quantified the degree of actin anisotropy in these conditions, and found the same trends. These results thereby confirm that the lack of response observed at very high loading rates is indeed governed by the stability of the actin cytoskeleton.

To clarify this issue, we have moved jasplakinolide results to figure 2, and explained them earlier in the manuscript.

2. The vimentin filaments are linked to paxillin-rich focal adhesion contacts, a region also rich in the motor protein myosin II. What is the role of intermediate filament network on loading rate, particularly in the fibroblast in vitro model? In the lung, the environment would be subjected to repeating loadings. It was recently shown (<https://doi.org/10.1073/pnas.1903890116>), that

the vimentin plays an essential role in maintaining the resilience of the cytoplasm because of its high yielding strain, while the rest of cytoplasmic components are greatly softened or even disassembled. If this is in fact what is happening how does an intact vimentin network effect the translocation of YAP and paxillin?

This is a very interesting question. As a preliminary analysis, we have now carried out vimentin stainings as a function of stretch (see figure below). Our images show that the vimentin cytoskeleton does appear to change organization with stretch: for cells either unstretched or stretched at low frequency, vimentin is primarily localized in the central part of the cell (compare with the tubulin image above, which shows the outline of the same cell). At the peak of the mechanosensing response (1 Hz), vimentin reaches further out into the cell periphery, where it may indeed interact with focal adhesions. The response at very high rates (square 1 Hz signal) is less clear. While this points at potentially very interesting phenomena, we feel that this lies out of the scope of the current manuscript, which aimed at establishing the role of force loading rate in actin-mediated mechanosensing. As the reviewer points out, vimentin has a high capacity for withstanding load. Thus, the intermediate filament network likely absorbs part of the load that would otherwise be transmitted to the actin cytoskeleton. This may indeed affect the forces and loading rates experienced by actin-linked mechanosensing events, like focal adhesion formation and also YAP translocation. Similarly, load bearing by vimentin is also likely to affect the fluidization response, which as stated above is a complex phenomenon that is bound to be tuned by any perturbation affecting not only the actin cytoskeleton but also its connection to intermediate filaments, and potentially also microtubules. However, we feel that analysing this in detail would require a full study on its own. To this end, we provide the images below merely as review material, and we would prefer at this stage not to include them in the publication.

Images of cells either not stretched (Ns) or stretched at the indicated frequencies with a triangular (Tr) or square (sq) signal. Scale bar, 40 μ m.

3. Figure 4: The *in vivo* approach shown alters frequency of ventilation and examines the impact on YAP translocation. High frequency ventilation (HFV) is a ventilatory strategy that utilizes a form of mechanical ventilation that typically combines very high respiratory rates (>60 breaths per minute) with tidal volumes that are smaller than the volume of anatomic dead

space. The clinical rationale for this type of ventilation is that gas exchange is optimized by utilizing small tidal volumes with minimal alveolar stretch.

To test the effect of varying the ventilation frequency on YAP, the authors applied a different ventilation frequencies and claimed that tidal volume was equal in each lung. The author will need to provide pressure-volume curves to support this claim.

We agree with the reviewer that, if applied ventilation had been pressure-controlled, potential differences in lung impedance could have led to different tidal volumes in both lungs. This would therefore require volume measurement as mentioned by the Reviewer. However, we would like to clarify that we applied volume-controlled differential ventilation. Accordingly, left and right lungs received the tidal volume specifically set in the syringe-based differential ventilator.

The left lung was ventilated at 0.1 Hz, and the right lung at 2.1 Hz. According to the authors, the animal received the same total minute ventilation, hence keeping O₂ and CO₂ blood gas levels thereby discarding any systemic effect induced by differential ventilation. It would be quite surprising that the right and left lungs had equal pO₂ and pCO₂ levels. The authors need to provide data demonstrating this important factor, particularly as YAP translocation to the nucleus has been shown to be oxygen dependent.

We thank the Reviewer for mentioning this issue, which we did not discuss in the manuscript. Indeed, as each lung was ventilated with different frequencies (and same tidal volume) the alveolar ventilation in each lung was different, and hence O₂ and CO₂ partial pressures in the alveolar gas of both lungs would be different. Such difference would obviously affect the epithelial and endothelial cells in the alveolar-capillary barrier. However, cells in the lung parenchyma are perfused with systemic oxygenated blood coming from the aorta by the bronchial circulation -also known as "pulmonary collateral circulation" or "systemic blood supply to the lungs" (Deffebach et al, Am Rev Respir Dis. 1987 Feb;135(2):463-81, 1987; Suresh and Shimoda, Compr Physiol. 2016;6(2):897-943, 2016). Therefore, given that we assessed YAP from whole-parenchymal lung sections, we can assume that most of those cells were perfused with systemic arterial blood, which was the same for both lungs. Anyway, we note that even in the unexpected case that cells in the low-frequency ventilated lung had lower O₂ and higher CO₂ levels than in the lung ventilated at high frequency, the expected effect on YAP would be the opposite to what we observe (see references 55,56, and 57 in the manuscript). Thus, even if present, changes in oxygen levels would not explain our results. We have now clarified this in page 9 of our manuscript.

Further, it should be mentioned that precisely measuring the local partial pressure along the lung parenchyma is virtually impossible even using the O₂ sensors equipped with the most miniaturized tip sensors (50 microns diameter). We have expertise in such oxygen measurement at local tissue level (doi: 10.1152/jappphysiol.00303.2019, doi: 10.5665/sleep.3848, doi: 10.5665/SLEEP.1176, doi: 10.1186/1465-9921-11-3, doi: 10.1183/09031936.00184314, doi: 10.5665/sleep.4166), and we think it is virtually impossible to place the sensor tip without physically disturbing the natural perfusion corresponding to pulmonary and bronchial circulation in rat lungs. Particularly taking into account that the lungs are not static (as other organs) but subjected to the intensive continuous deformation because of ventilation.

Additionally, the authors should consider measuring alveolar deformation with the given ventilation volumes and frequencies. Ensuring that each alveolar deformation is similar in magnitudes would allow the authors to link (or not) that YAP nuclear localization is due to stretch alone.

In connection with our response to the previous question on lung tidal volumes, given that we precisely set the same tidal volume for right and left lungs, we may very reasonably assume that alveolar deformation was the same in both lungs. Carrying out these measurements would require using a very complex methodology such as intravital lung microscopy (doi: 10.1117/1.JBO.20.6.066009). And it is not clear whether this technique could be applied to measure the undisturbed deformation of alveoli breathing at a frequency as high as 2.1 Hz.

The authors state that in Figure 4, the cells in the alveoli are mostly epithelial and endothelial cells. In contrast, the prior figures use fibroblast; different cell types may respond differently to loading rate. It would be helpful if the authors co-immunostained and determined which cells in the lung had YAP translocate to the nucleus.

As suggested by the reviewer, we attempted to co-immunostain cell type markers and YAP in rat lung samples. However, we encountered problems in unambiguously determining cell types due to the limited fluorescence channels available. As an alternative, we repeated stretch experiments in vitro in lung epithelial and endothelial cells. Our results show that both cell types exhibit the same trends than fibroblasts, with an initial increase in YAP ratios and paxillin focal adhesion lengths with frequency, and then a decrease. Interestingly, the point at which the decrease begins depends on cell type, happening at 2 Hz in both fibroblasts and endothelial cells, but only for the 1 Hz square signal (with a loading rate a bit above that corresponding to the triangular 2 Hz signal) in the case of epithelial cells. Thus, our described response seems to be generalizable to different cell types, but the thresholds at which the different regimes occur are cell-type dependent. This is consistent with previous findings showing that mechanosensing, as predicted by the molecular clutch model, can be tuned by differences in integrin expression or availability, myosin contractility, and other parameters (Elosegui-Artola et al., 2014 doi.org/10.1038/nmat3960 and 2016, doi.org/10.1038/ncb3336, Oria et al. 2017, doi.org/10.1038/nature24662), which are bound to depend on cell type. We now show these new data in supplementary fig. 3.

Additionally, it would be helpful to have link between the in vitro and in vivo experiments (e.g. same cell type). This is important because different rates of cyclic stretch can cause cell proliferation and death (Liu et al., AJP, 1992; Tschumperlin et al., AJRCCM, 2000). Given this information, it would be useful to know the specific functional outcomes (proliferation, apoptosis, collagen production) of force loading rate on cells both in vitro and in vivo.

In vivo, our differential lung ventilation experiment in rats was limited to 1 h to minimize the potential risk of ventilator-induced lung injury, which increases with the duration of mechanical ventilation (Marini et al. 2020, doi:10.1164/rccm.201908-1545CI). Unfortunately, this experiment duration was not sufficiently long to clearly assess potential differences in cell responses. In vitro, however, we were able to measure how the different stretch conditions affected proliferation, which is one of the main known outcomes of mechanosensing. Of note, observing proliferation outcomes indeed required a time of stretch of 2.5 hours, well above the window of 1 h of in vivo experiments. The results showed that proliferation rates (as assessed with an EdU incorporation assay) mimicked the trends observed in mechanosensing,

i.e. an initial increase with loading rate followed by a subsequent decrease. The results are shown in supplementary fig. 1c.

Minor comments:

The authors need to report ANOVA test statistics; it is unclear what is significant by looking at the figures and figure captions.

We have now included additional ANOVA test statistics in the legends of figures 1, 4, S1, S4, and S5 for all relevant panels.

Not formatted in journal style (just "Main"; not "Introduction," "Results," "Discussion," etc.)

We have now formatted the text in journal style.

Reviewer #3 (Remarks to the Author):

In this manuscript, González-Tarragó et al. investigate how the force loading rate impacts cellular mechanotransduction. This is an important yet underexplored aspect of cell biology, and thus the current study is very timely. Using a number of different complementary experimental approaches, the authors convincingly demonstrate a force rate dependency of mechanotransduction and connect it to fluidization/damage of force-transmitting proteins in cells (see below). The experiments are elegant, the paper is well written, and most of the conclusions are justified. The data reveal a new mechanotransduction mechanism that will be highly relevant for many future studies in different fields.

There are two major points, however, which the authors should address.

1) I am not sure if what the authors observed really corresponds to a fluidization of cells. If I understand it correctly, all observations can simply be explained by structural damage (or plastic deformation after the yield point), which is, technically speaking, not the same as fluidization. Fluidization might be a consequence of structural damage, but it is not necessarily the cause of the observed behavior. It's just a terminology issue, but an important one. As fluidization was not shown directly anyway (e.g., a change in the cells' viscosity), I'd strongly recommend to re-word the manuscript and avoid using the term 'fluidization'.

We fully agree with the reviewer that fluidization may not be the most accurate term. In its original definition (for instance in Trepatt et al., and Kollmannsberger and Fabry, doi.org/10.1146/annurev-matsci-062910-100351) fluidization was meant to describe both cytoskeletal softening and, as the reviewer points out, a more viscous-like behaviour, hence its name. However, it has later been used widely to refer simply to the well-known phenomenon of cytoskeletal softening upon stretch (see for instance Harris et al., doi.org/10.1073/pnas.1213301109, or Nava et al., doi.org/10.1016/j.cell.2020.03.052). In our case, we are also simply referring to cytoskeletal softening. Although we acknowledge the imprecision, we have preferred to keep this term in the manuscript to best link it to related literature. However, we have clarified this potential confusion in the text, and defined our interpretation of the term precisely (page 5).

2) I have some questions about the analysis of the AFM experiments. The authors currently simply fit a linear function through 2 fixed points on a highly non-linear curve, concluding that cells liquify (or better: likely undergo structural damage) at a loading rate between 5 and 6 $\mu\text{m/s}$. Looking at the raw data in Fig. 3b, however, I would argue that large parts of the slopes in these two plots (the more reliable parts further way from the point of detachment) look nearly identical, and that the fits are actually rather poor. How would the stiffness distributions look like if only data above 0.5 nN (or a similar threshold) would be analysed? And how if a nonlinear model, such as the Hertz model, would be used to analyse the data? Would Fig 3b still suggest a similar tendency? If not, the authors should re-think their conclusions.

We apologize since this experiment was not clearly explained in the manuscript. Because the aim of the experiment is to assess whether stretching cells leads to their softening, the relevant part of the curves is in fact the part of negative forces, in which the cell is being stretched. Indeed, in the initial part of the curves in fig. 4b and 4e with positive forces, the cells remains under compression. Therefore, restricting our analysis to the part of the curves with forces above 0.5 nN, as suggested by the reviewer, would mean that we are only analysing cells under

compression, which was not our aim. Instead and to improve our analysis, we have now restricted the analysis to the part with negative forces. Further and as suggested by the reviewer, we have used a DMT model, which is a modification of the Hertz contact model to take adhesion into account. Essentially, the curve is fitted to a Hertz contact model, but with the modification that the curve is offset by the maximum pulling force (force of adhesion). That is and taking the curves in fig. 4b as an example, we fitted the curve starting from zero force and ending in the maximum pulling force. Later time points, in which the cell starts detaching and force progressively goes back to zero, were also excluded. The trends that we reported previously are all maintained, and in fact they are now even clearer. We have now clarified this in the figure itself and in the methods.

Furthermore, the authors should provide information about what the data points show (mean +- SEM?) and about the statistical analysis in the figure caption. Assuming that they used a multiple comparisons test in Fig. 3c as mentioned, amongst other tests, in the methods, is the difference between 5 and 6 $\mu\text{m/s}$ the only one that is statistically significant? So is there no difference between 2 and 4 and 8 $\mu\text{m/s}$, for example? What does that mean?

All data points in the figures are shown as mean +- SEM, as indicated in the methods. Our aim in this experiment was to evaluate if progressively increasing pulling speed eventually led to a softening event. Because such an event was specifically observed between 5 and 6 $\mu\text{m/s}$, we specifically compared only these two points with a non-parametric equivalent of a paired t-test (Wilcoxon matched-pairs signed rank test). After improving our analysis as described in response to the comment above, the p-value of this comparison is $p=0.0042$. To be more rigorous, instead of a t-test we have now carried out a one-way non-parametric ANOVA for all conditions (Friedman Test), and then within that analysis tested for the specific difference between 5 and 6 $\mu\text{m/s}$. Such an analysis shows an overall significance of the effect of pulling speed ($p<0.0001$), and also significance specifically of the 5-6 $\mu\text{m/s}$ comparison ($p=0.0235$). Of course, we could test for additional comparisons as the reviewer suggests, but these do not respond to any question or hypothesis in our manuscript, so we don't think they would add relevant information to our analysis. We have now clarified this in the figure caption.

Minor points:

Abstract: I would omit 'In contrast' in sentence 4, as it could well be in addition.

We have omitted "in contrast" as suggested.

Main text: 'the fundamental mechanical variables that cells sense and respond to are unknown'. I would disagree with this statement and suggest to change it to ... not fully understood.

We have corrected this as suggested.

Page 4 paragraph 1: 'applying a very mild stretch (2.5% biaxial stretch, applied cyclically with a triangular 0.125 Hz wave for 1 h) leads to a deformation speed of ~ 60 nm/s, of the same order of magnitude than internally generated actomyosin flows.' It is confusing to compare these two time scales here, as they are not related (or, if they are, then rather inversely): An increase in

deformation speed leads to more mechanotransduction, while an increase in actin flow rates generally leads to less mechanotransduction. I'd suggest to omit the second part of this sentence.

This is a very interesting point, which is in fact at the core of the message of our work. Indeed, the reviewer is correct that increased actin flow rates are generally associated with low mechanotransduction. This is because actin flows and mechanotransduction are strongly coupled: as mechanotransduction occurs and focal adhesions form, they provide a stronger attachment to actin, slowing actin flows down. However, before this coupling occurs, one would expect actin flows to be of the same order regardless of the mechanical environment. This is indeed the case if one compares flows in cells on soft substrates (with very weak adhesions) and on the lamellipodia of stiff substrates, where adhesions have not yet matured (Fig. 1a,b). This concept is at the foundation of the molecular clutch theory: as a given actin flow engages through integrins to substrates with different mechanical properties (for instance, increased stiffness), loading rates will increase, and this will drive mechanotransduction. In turn, this will slow down actin flows until reaching a balance.

Precisely due to this coupling, it is not possible to independently assess the role of loading rates (or deformation rates) just by relying on the forces and deformations generated by cells themselves. Instead, externally imposed deformations, unaffected by any coupling, must be applied. This is precisely what we did here: we started by a deformation rate (2.5% biaxial stretch, applied cyclically with a triangular 0.125 Hz wave) which is comparable to internally generated actomyosin flows in low adhesion conditions (i.e., before they become coupled to adhesions). Then, we progressively increased this, to see whether we could reproduce the expected effects of loading rates. To be more precise, we have now specified “in low adhesion conditions” in the sentence mentioned by the reviewer.

Page 6 paragraph 1: 'Thus, at high levels of stretch or stretch rates, cytoskeletal softening would reduce the loading rates being applied to adhesions, since loading rates depend not only on the rate of deformation but also on the stiffness of the structure being deformed.' I agree with this statement, but wouldn't this also lead to a change in load amplitude, which might be partly contributing to what the authors see as well? The authors might briefly want to discuss this.

We agree with the reviewer, and we have modified this accordingly (page 9).

It would be nice to add a paragraph to the discussion about how the loading rate may impact structural damage within cells.

To address this and in response to concerns by other reviewers, we have now developed a computational clutch model. Based on our previous work (see Elosegui-Artola et al., Nat. Cell Biol. 2016), this model considers progressive force application to links between actin, talin, integrins, and fibronectin i.e., “clutches”). Then, it considers how talin unfolding and integrin-fibronectin unbinding depend on force (based on experimental single molecule data). As force builds in each timestep, the monte-carlo simulation evaluates whether talin will unfold, or integrins will unbind. If talin unfolds before the integrin unbinds, we assume that there is a mechanosensing event, which leads to integrin recruitment (i.e., adhesion growth in experiments). As a modification from our previous model, here we introduce that i) force on clutches does not arise from actomyosin contractility, but from an externally imposed stretch, and ii) the clutch can be disengaged not only by integrin unbinding, but also by actin cytoskeleton disruption (i.e., fluidization) above a threshold force. Of note, this event in the

model cannot distinguish between different potential events, such as breaking of actin filaments, or severing of actin crosslinks, for instance. However, the model provided a good fit to the data by assuming a force of about 140 pN, in reasonable agreement with reported experimental values for the breaking of actin filaments (Kishino and Yanagida, Nature 334, 74–76, 1988). Importantly, the model does not assume any dependence on loading rate per se. Simply, if loading rates are low, forces stay long enough in a low regime where integrin unbinding is more likely than talin unfolding. If loading rates increase, forces quickly reach a regime in which talin unfolding is more likely, triggering mechanosensing. For very high loading rates, the force required for fluidization is reached before talin unfolding can occur. In the case of optical tweezers experiments, fluidization is prevented for two reasons. First, tweezers are limited to forces below 100 pN. Second, measurements are carried out in the lamellar region, with an actin network much more structured than on the rounded cell phenotype found on soft substrates before stretch. Thus, forces applied to beads are likely distributed among many filaments, likely reducing the likelihood of fluidization. Consistently with this, loading-rate induced softening in AFM experiments (indicative of fluidization) is observed when rounded cells are pulled, but not when beads attached to lamellipodia are pulled. In the case of lungs, and although it is hard to assess in detail due to the much less controlled nature of the setup, it is tempting to speculate that tissue architecture has built-in mechanisms to distribute loads and prevent disruption of the actin cytoskeleton.

The model and its interpretation are now introduced throughout the text, and its results are shown in figure 2.

Statistical analyses should also be conducted for supplementary figures. If they were done and no differences observed, this should be stated as well.

We note that the supplementary figures already contained different statistical analyses, but we agree with the reviewer that some extra analyses could be conducted for further clarity. We have carried out additional statistical analyses in supplementary figures 1, 4, and 5, which we report in the legends.

Reviewer #4 (Remarks to the Author):

The authors indicate that the loading rate of force application is a key driver of mechanosensing. However, above certain thresholds the cytoskeleton collapses/fluidizes/softens, and this prevents strong adhesion to the ECM, as focal adhesions and the cytoskeleton lack reinforcement. In isolated rat lungs ex vivo, they demonstrate the relevance of the loading rate (but not of fluidization). The key message here is that what cells read are force dynamics, rather than absolute values or thresholds.

I have few, mainly conceptual questions, some requiring additional experiments, and other points just discussion:

1) They assume (line 74) that cell-substrate attachment and force transmission occurs largely at the cell periphery. Is it so? A recent paper by Vogel and colleagues (Shiu et al 2018) has instead shown that the central (perinuclear) area of the cell displays the more robust traction force, and is mainly responsible for YAP nuclear accumulation. How is this changing their conclusions?

This is a very interesting point. In our stainings, focal adhesions were generally more prominent at the cell periphery (see for instance paxillin images in figure 1). Since focal adhesions were one of the main readouts that we used for mechanotransduction, and also correspond to a structure expected to be under force, we think that estimating the size of cells by taking the cell periphery as a reference is appropriate. However, we note that this was only used to estimate a typical size of cells, and thereby the magnitude of deformation speeds imposed by stretch. This was done simply to show that deformations mediated by stretch, and those mediated by actomyosin flows were of the same order of magnitude. Considering the central zone rather than the periphery would change the size and speed estimate only by approximately 2-fold, the reported orders would remain in the same range.

To clarify this, we have now specified that we take the cell periphery as a reference due to the location of focal adhesions (page 4).

2) They show the relevance of the loading rate, either by increasing ECM stiffness, or by increasing the frequency of deformations using stretching pulses on cells with a stretching device. For example they show that the same effect on YAP or Focal adhesion (FA) can be obtained either by 4x stiffness (E) or by keeping E constant (of a soft ECM) and increasing the frequency from 0.125 to 1 Hz. This is certainly interesting and intriguing. But how does it work? In the revised molecular clutch model (Elosegui-Artola 2016) the key determinant of mechanosensitivity is whether the resistive force of the ECM can be transmitted to talin before the integrin-ECM bond dissociates. In that model, on a soft ECM, integrins dissociate from their ECM ligands faster than (and thus before) any engagement of actin to integrin (through clutching molecules). Here: How is stretching rate and frequency changing that? Are these stabilizing/increasing the binding affinity/avidity of integrin for ECM? or are these inputs making molecules such as talin or vinculin loading at faster frequency (compared to the control "still" soft ECM) on integrin, leading to reinforcement? The authors should offer some explanations or interpretations.

To address this and in response to concerns by other reviewers, we have now developed a computational clutch model to interpret our results. Based on our previous work (see Elosegui-

Artola et al., Nat. Cell Biol. 2016), this model considers progressive force application to links between actin, talin, integrins, and fibronectin i.e., “clutches”). Then, it considers how talin unfolding and integrin-fibronectin unbinding depend on force (based on experimental single molecule data). As force builds in each timestep, the monte-carlo simulation evaluates whether talin will unfold, or integrins will unbind. If talin unfolds before the integrin unbinds, we assume that there is a mechanosensing event, which leads to integrin recruitment (i.e., adhesion growth in experiments). As a modification from our previous model, here we introduce that i) force on clutches does not arise from actomyosin contractility, but from an externally imposed stretch, and ii) the clutch can be disengaged not only by integrin unbinding, but also by actin cytoskeleton disruption (i.e., fluidization) above a threshold force. Of note, this event in the model cannot distinguish between different potential events, such as breaking of actin filaments, or severing of actin crosslinks, for instance. However, the model provided a good fit to the data by assuming a force of about 140 pN, in reasonable agreement with reported experimental values for the breaking of actin filaments (Kishino and Yanagida, Nature 334, 74–76, 1988). Importantly, the model does not assume any dependence on loading rate per se. Simply, if loading rates are low, forces stay long enough in a low regime where integrin unbinding is more likely than talin unfolding. If loading rates increase, forces quickly reach a regime in which talin unfolding is more likely, triggering mechanosensing. For very high loading rates, the force required for fluidization is reached before talin unfolding can occur.

We note that the underlying factor in the model driving the response to altered mechanical parameters is the same here, and in our previous Elosegui-Artola 2016 work: loading rates. Here, loading rates are increased by frequency and amplitude. In Elosegui-Artola et al., increased loading rates are achieved by making the substrate stiffer.

We now present the model throughout the text (see detailed description in methods), and its predictions are shown in figure 2.

3) Is this connected to cell shape, such that a cell that is challenged to spread by stretching pulses compatible with the lifetime of integrin-ECM bonds responds by increasing its own pulling, with faster clutch loading, with talin deformation and vinculin-mediated reinforcement, as it occurs in cells on stiffer ECM? This is not shown, but could be interesting to demonstrate on cells experiencing the stretching device, and not only using beads in Fig2.

As the reviewer suggests and as explained in response to the comment above, we believe that the mechanism is indeed the same in response to increased stretch frequencies, and increased stiffness. Of note, the lower stretch frequencies used in our experiments (0.125 Hz, period of 8 s) are already of the order of integrin lifetimes (also of the order of seconds, see for instance Kong et al., JCB 2009). At these slow timescales, the model predicts (and experiments show) that focal adhesions should not form, because integrin unbinding would occur before talin unfolding.

Regarding pulling by the cell itself, we agree with the reviewer that this is most likely occurring. However, the order of relative cell-substrate deformation speeds induced by stretch is only comparable to internal actomyosin flow speeds at the mildest stretch condition assessed (0.125 Hz, 2.5% stretch). In all other conditions, deformation and loading rates imposed by stretch are much larger, and can thus be safely assumed to dominate our observed responses. This is now clarified in page 4 of the manuscript.

4) The simplest interpretation is that mechanosensing occurs only above (absolute?) thresholds that are or not reached depending on for how long/how often cells experience the maximal amount of (tolerated) stretch in a given time window. Square stretches gain their target before and more robustly than triangular ones, and increasing E further facilitates the goal. As if cells are able to integrate all these inputs and count them, accumulating them. Please comment.

We believe that these interpretations would not be consistent with the complexity of our results:

- If mechanosensing was driven by the amount of time (how long) cells spend above a given value of stretch, then decreasing frequencies should increase mechanosensing responses. Indeed, cells stretched at the lowest frequencies would spend longer uninterrupted lengths of time above a given threshold. However, this is the opposite trend to what we observe. Also, square signals should always increase the response, which is not what we observe.
- If mechanosensing was driven by how often cells surpass a given value of stretch, then square and triangular signals should have the same responses, which is not the case.
- Of course, mechanosensing could be driven by a combination of how long/how often a specific signal is used, but then validating or disproving this hypothesis would require predicting exactly how the two factors should be combined. In fact, we believe that our introduced molecular clutch model does precisely that, by taking into account known properties of the molecular players involved.

5) they also report that at very high stretch rates/ stretch magnitudes cells lose the connection with their ECM, as this causes softening of their cytoskeleton. My concern here is why is this not happening on cells that are "normally" attached to glass or plastic? Is this connected to the need of some minimal and required timing in the dynamic of FA function - globally, at the cell level - that once overruled by experimental stretching leads to failure of all clutches and loss of traction?

This is a highly relevant question, and in fact there is evidence that this does occur in cells attached to very stiff substrates. Indeed, cells seeded on glass substrates with a very limited availability of ECM ligands fail to form focal adhesions and localize Yap to the nucleus (see for instance Arnold et al. *ChemPhysChem* 5, 383–388, 2004). In previous work, we showed that cell sensing of ECM ligand density can be explained by a role of the loading rate, as predicted by a molecular clutch model (Oria, R. et al., *Nature* 552, 219–224, 2017). Decreasing the available density of ECM ligands means force applied by cells is distributed among less ligands, increasing applied force (and loading rate) per ligand. We showed that as ECM density decreases and loading rate increases, there is a biphasic response of focal adhesions and YAP, similarly to what we observed here with frequency.

In this previous work, the role of the loading rate was merely inferred from the model and not controlled directly, and the failure to form adhesions at very sparse ligand densities was interpreted through a generic saturation of adhesions. However, the results are fully consistent with fluidization at high loading rate. Thus, we hypothesize that in typical conditions in glass, even if the substrate is very stiff, cell contractility is distributed among enough ligands to prevent fluidization. However, if ECM ligand density is decreased sufficiently, this regime becomes apparent.

We have included a reference to this in the discussion of the paper, page 10.

6) At the end, to what extent is this cell relaxation phenomenon really physiological? It is potentially intriguing, yet the lung ventilation experiments do not show any fluidization, for example, but only increased response to increased stretching frequency. What about other cells and tissues with much faster stretching and relaxation rates, such as heart or vessels?

We have now carried out different experiments that support the potential physiological relevance of our findings. First, we have seen that the changes in mechanosensing that we observe are mirrored by changes in proliferation, a major outcome of mechanosensing responses (Supplementary fig. 1c). Second, we have determined that the biphasic response to stretch observed applies not only to fibroblasts, but also to endothelial and epithelial cells (Supplementary fig. 3). However, the reviewer is correct that in our in vivo experiments in lungs only the phase of increased response was observed. We speculate that this may be due to large scale tissue-level protection mechanisms taking place in the lung, but this remains to be studied. As the reviewer points out, the regime of cytoskeletal fluidization, may be more easily observed in faster stretching in vivo scenarios, such as the heart, or even vocal cords. Unfortunately, exploring this would require different entire studies of their own, and is not doable within the scope of this manuscript. It is nevertheless a very interesting discussion, and we have included it at the end of our discussion section (page 10).

7) They point at actin filaments to explain fluidization. But what about microtubules or intermediate filaments?

This is a very interesting question, and to assess potential changes in the organization of intermediate filaments and microtubules, we have now carried out stainings as a function of stretch. Our images suggest that particularly the microtubule cytoskeleton may be altered in the conditions where actin is also more organized (triangular 1 Hz signal), confirming the concept that the networks are linked. Whereas our results show a clear role of the actin cytoskeleton, this of course does not discard that the coupling to intermediate filaments and microtubules is important. However, we feel that analysing this in detail would require a full study on its own, and lies out of the scope of this manuscript. To this end, we provide the images below merely as review material, and we would prefer at this stage not to include them in the publication.

Images of cells either not stretched (Ns) or stretched at the indicated frequencies with a triangular (Tr) or square (sq) signal. Scale bar, 40 μm .

REVIEWERS' COMMENTS

Reviewer #1 (Remarks to the Author):

In general, the authors have done a fine job of responding to my concerns. The authors are now much clearer about what they mean by “fluidization” and the new clutch model adds some new mechanistic insight into the rate-dependence of that fluidization. I also find the explanation of the discrepancies between optical tweezer and substrate stretch experiments convincing. Finally, the new frequency-dependent proliferation results, while not necessarily attributable only to YAP, at least show that the dynamic range of YAP N/C ratios is phenotypically significant. I have two remaining comments:

1. Given its mechanistic importance, the new model could use a bit of clarification. For example, the authors initially describe the model action as a Monte Carlo simulation (typically associated with generation of distributions of configurations through a series of randomized perturbations) but then later frame it as more of a dynamics-style simulation in which time-dependent trajectories are calculated. Are there elements of both types of models here? Also, with the large number of model parameters in play, it would be helpful to have some sense of the robustness of the model. For example, how sensitive are the predictions in Fig 2 to the choices of integrin density, off-rates, etc.
2. The authors openly acknowledge that “fluidization” in this study really means “cytoskeletal softening” (lines 121-123) and is not intended to imply anything about viscous vs elastic behavior. In my view, fluidization isn't fluidization with changes in viscous properties, and the authors' imprecise use of the term is likely to be a major source of confusion to readers (as it was here for multiple reviewers). The argument about needing to maintain consistency with other papers seems unpersuasive and advocates for propagating an error. I would therefore urge the authors to write more plainly and use the term “softening” or something similar, especially in the title. Precision on this point is especially important with the field increasingly investigating the biological importance of viscous/loss properties of cells and biomaterials.

Reviewer #2 (Remarks to the Author):

Reviewer #2 (Remarks to the Author):

Main Criticisms

1. The revised definition of fluidization is nicely presented in the new version of the manuscript and clears any lingering confusion. The addition of the computational model strengthens the impact of the manuscript, and it seems to fit the data in Figure 2 robustly. I would be interested to see whether the model fits data collected using other cell types, such as those presented in Supplementary Figure 3. Still, I appreciate that this may be outside the scope of the current manuscript.
2. I agree with the authors that the jasplakinolide experiments are better suited for Figure 2. However, we echo our initial concerns surrounding the blebbistatin experiments. While the authors show that blebbistatin treatment abrogates mechanosensing in the stretch, but not the optical

tweezer experiments, I believe that their conclusions are overstated. The authors never show a decrease in cell stiffness with blebbistatin treatment. I recommend that the authors change the language in the text and use the discussion to explore the possible mechanical and chemical signaling effects that blebbistatin treatment might cause.

3. I appreciate the new data the authors have presented in their responses. However, the critical issue that was to be address—What is the effect of loading rate on the cytoskeleton? The cytoskeleton is composed of microfilaments (actin), intermediate filaments, and microtubules. As the authors re-iterate, vimentin has a high capacity for withstanding load. Thus, the intermediate filament network likely absorbs part of the load that would otherwise be transmitted to the actin cytoskeleton. This may indeed affect the forces and loading rates experienced by actin-linked mechanosensing events, like focal adhesion formation and also YAP translocation. Similarly, load-bearing by vimentin is also likely to affect the fluidization response, which, as stated above, is a complex phenomenon that is bound to be tuned by any perturbation affecting not only the actin cytoskeleton but also its connection to intermediate filaments, and potentially also microtubules. I agree that performing experiments on intermediate filaments may be outside the scope of this paper. Thus, the language in the text needs to be more precise to reflect the study of the actin cytoskeleton and not the cytoskeleton as a whole. Also, the discussion should address the potential role of the intermediate filament network in contributing to the loading rates and YAP translocation.

4. Ref 55-57 are insufficient to support the claim that lower O₂ is expected to result in higher YAP nuclear localization, as these experiments were conducted in organs that have significantly lower pO₂ levels than the lungs at homeostasis. Also, the authors neglect to mention what might happen in a hypercapnic environment. I appreciate the difficulty of precisely measuring the local partial pressure along the lung parenchyma, but a reasonable proxy would be blood gas measurements. I recommend probing O₂ by ventilating both lungs at the same frequency and measuring pO₂. By comparing O₂ levels at different frequencies, the authors should identify whether pO₂ levels are genuinely equal. Alternatively, the authors could ventilate the lungs with hypoxic, normoxic, and hyperoxic O₂ levels and examine YAP nuclear localization.

5. The response provided to address whether or not the cell in the previous Figure 4 are primarily epithelial and endothelial cells is not sufficient. I appreciate the response and find it interesting that the biphasic mechanosensing response is conserved across different cell types in vitro, albeit at different frequency thresholds. However, the need for co-immunostaining remains essential, as it is difficult to interpret the in vivo responses without resolving the cell types that undergo mechanosensing. I recommend referencing doi: 10.1172/JCI125014 for co-immunostaining experiments.

Reviewer #3 (Remarks to the Author):

The authors have carefully addressed the reviewers' comments and incorporated new aspects into the manuscript. Particularly the addition of the new clutch model is highly valuable and explains the data nicely.

I also appreciate that the authors have attempted to briefly clarify what they mean when they refer to fluidization of cells. However, I still find this term confusing (as did two other reviewers), and

there is no experimental data supporting a fluidization of cells. The authors either need to provide experimental proof of a change in the cells' fluidity (which I don't think is necessary as it is not required to support the main findings of this study) or revise their wording more extensively. Forces above a threshold are likely to disrupt the cytoskeleton (including actin), which is supported by the new data presented by the authors, and this disruption may or may not lead to a cell's fluidization. But fluidization is not shown. Hence, I would strongly suggest talking about cytoskeletal disruption rather than about fluidization. The casual use of the word in earlier papers doesn't justify its usage here, and in the initial papers, as the authors correctly pointed out, viscosity had been measured to show a change in fluidity.

Particularly, I would suggest to replace the word 'fluidization' by 'cytoskeletal disruption' or something similar in the title and figure captions, to change the abstract accordingly (for example to: 'However, above a given threshold the actin cytoskeleton is disrupted and softens, decreasing loading rates and preventing reinforcement. '), and to introduce the idea of fluidization only in the discussion, as it is one possible interpretation of the data.

Minor points:

I would suggest adding the DAPI images to Fig. 1c.

Add enlarged images of the blue squares (fits) in Fig. 4b & e.

Reviewer #4 (Remarks to the Author):

The authors have addressed all my comments and have done a good job at addressing the overall key concerns that have emerged throughout the revision process. I keep supporting publication of this work. This is a nice demonstration of a multidisciplinary effort combining classic read outs with YAP nuclear/cytoplasmic localization as proximal and functionally valid read-out in mechanosignaling studies. In this view, it may be appropriate and useful for the lay reader to add at lines 57 or 68 a review that cover this concept more broadly than the sole Dupont et al., 2011 (e.g., Brusatin et al., Nat Mater 2018)

Reviewer #1 (Remarks to the Author):

In general, the authors have done a fine job of responding to my concerns. The authors are now much clearer about what they mean by “fluidization” and the new clutch model adds some new mechanistic insight into the rate-dependence of that fluidization. I also find the explanation of the discrepancies between optical tweezer and substrate stretch experiments convincing. Finally, the new frequency-dependent proliferation results, while not necessarily attributable only to YAP, at least show that the dynamic range of YAP N/C ratios is phenotypically significant. I have two remaining comments:

We thank the reviewer for the positive assessment of our manuscript and revisions.

1. Given its mechanistic importance, the new model could use a bit of clarification. For example, the authors initially describe the model action as a Monte Carlo simulation (typically associated with generation of distributions of configurations through a series of randomized perturbations) but then later frame it as more of a dynamics-style simulation in which time-dependent trajectories are calculated. Are there elements of both types of models here?

We refer to the model as a Monte Carlo simulation in the sense that it contains stochastic elements. That is, at each time step, the different molecular events that can occur (such as integrin binding and unbinding, or talin unfolding) are decided stochastically, according to the probability distributions given by their respective rates. However, as the reviewer points out this is indeed done for each time step, and thus as a function of time, leading to time-dependent trajectories. We note that this is the same approach that we used in several previous implementations of this model (see Elosegui-Artola et al., *nat. mater.* 2014, Elosegui-Artola et al., *Nat Cell Biol* 2016, Oria et al., *Nature* 2017). In turn, all of our models were based on previous models by the group of David Odde (Chan and Odde, *Science* 2008) which used the same approach. To be more specific, we have now described our model as a stochastic simulation (rather than Monte Carlo) in the methods section (line 594).

Also, with the large number of model parameters in play, it would be helpful to have some sense of the robustness of the model. For example, how sensitive are the predictions in Fig 2 to the choices of integrin density, off-rates, etc.

Although the model does have many parameters, we note that the majority of them were taken directly from our previous work (Elosegui-Artola et al., *Nat. Cell Biol.* 2016) and were not adjusted or modified here (see supplementary table 4). Indeed, our current model is essentially the same as in that paper, by simply adapting it to deformations imposed externally rather than by actomyosin contraction. Regarding sensitivity, we refer the reviewer precisely to the sensitivity study we did in our Elosegui-Artola 2016 paper, where this was assessed for the different parameters (supplementary table 1 of the paper). We have now specified this reference for the sensitivity analysis in the methods (line 708).

2. The authors openly acknowledge that “fluidization” in this study really means “cytoskeletal softening” (lines 121-123) and is not intended to imply anything about viscous vs elastic behavior. In my view, fluidization isn't fluidization with changes in viscous properties, and the authors' imprecise use of the term is likely to be a major source of confusion to readers (as it was here for multiple reviewers). The argument about needing to maintain consistency with

other papers seems unpersuasive and advocates for propagating an error. I would therefore urge the authors to write more plainly and use the term “softening” or something similar, especially in the title. Precision on this point is especially important with the field increasingly investigating the biological importance of viscous/loss properties of cells and biomaterials.

To increase precision and as suggested by the reviewer, we have changed the term to “cytoskeletal softening” throughout the text, including the title.

Reviewer #2 (Remarks to the Author):

Reviewer #2 (Remarks to the Author):

Main Criticisms

1. The revised definition of fluidization is nicely presented in the new version of the manuscript and clears any lingering confusion. The addition of the computational model strengthens the impact of the manuscript, and it seems to fit the data in Figure 2 robustly. I would be interested to see whether the model fits data collected using other cell types, such as those presented in Supplementary Figure 3. Still, I appreciate that this may be outside the scope of the current manuscript.

We thank the reviewer for the positive assessment.

2. I agree with the authors that the jasplakinolide experiments are better suited for Figure 2. However, we echo our initial concerns surrounding the blebbistatin experiments. While the authors show that blebbistatin treatment abrogates mechanosensing in the stretch, but not the optical tweezer experiments, I believe that their conclusions are overstated. The authors never show a decrease in cell stiffness with blebbistatin treatment. I recommend that the authors change the language in the text and use the discussion to explore the possible mechanical and chemical signaling effects that blebbistatin treatment might cause.

We agree with the reviewer that myosin inhibition may have complex effects beyond the changes in actin cytoskeletal structure that we described. We have now modified the text to acknowledge that, whereas the effects of blebbistatin treatment are consistent with our hypothesis, other effects of the treatment may be at play (text starting in line 260).

3. I appreciate the new data the authors have presented in their responses. However, the critical issue that was to be address—What is the effect of loading rate on the cytoskeleton? The cytoskeleton is composed of microfilaments (actin), intermediate filaments, and microtubules. As the authors re-iterate, vimentin has a high capacity for withstanding load. Thus, the intermediate filament network likely absorbs part of the load that would otherwise be transmitted to the actin cytoskeleton. This may indeed affect the forces and loading rates experienced by actin-linked mechanosensing events, like focal adhesion formation and also YAP translocation. Similarly, load-bearing by vimentin is also likely to affect the fluidization response, which, as stated above, is a complex phenomenon that is bound to be tuned by any perturbation affecting not only the actin cytoskeleton but also its connection to intermediate filaments, and potentially also microtubules. I agree that performing

experiments on intermediate filaments may be outside the scope of this paper. Thus, the language in the text needs to be more precise to reflect the study of the actin cytoskeleton and not the cytoskeleton as a whole. Also, the discussion should address the potential role of the intermediate filament network in contributing to the loading rates and YAP translocation.

As requested, we have corrected the text throughout to specify when we refer specifically to the actin cytoskeleton. We have also added a discussion of the potential effects of intermediate filaments (text starting in line 306), which we agree are very interesting and merit further work.

4. Ref 55-57 are insufficient to support the claim that lower O₂ is expected to result in higher YAP nuclear localization, as these experiments were conducted in organs that have significantly lower pO₂ levels than the lungs at homeostasis. Also, the authors neglect to mention what might happen in a hypercapnic environment. I appreciate the difficulty of precisely measuring the local partial pressure along the lung parenchyma, but a reasonable proxy would be blood gas measurements. I recommend probing O₂ by ventilating both lungs at the same frequency and measuring pO₂. By comparing O₂ levels at different frequencies, the authors should identify whether pO₂ levels are genuinely equal. Alternatively, the authors could ventilate the lungs with hypoxic, normoxic, and hyperoxic O₂ levels and examine YAP nuclear localization.

We note that in animal experiments, the reduction of ventilatory frequency in one lung was compensated with increased frequency in the other lung. Therefore, ventilation in the animals was kept constant in all cases (i.e., there is no condition in which both lungs were ventilated at low or high frequencies). We apologize because after reviewing the text we have realized that this may not have been fully clear, and we have now revised the text to clarify it (text starting in line 270). Thus, the proposed experiments of measuring pO₂ after ventilating both lungs at the same frequency would not correspond to any of the experimental conditions used in our study, and would not be useful. Further, considering that the cells in the lung parenchyma are perfused with systemic oxygenated blood from both lungs, the oxygen and CO₂ levels should be maintained. Finally and although there are no studies carried out in pulmonary cells in response to hypoxia, all studies carried out in several other tissues show that a reduction of oxygen levels respect to their physioxenic values results in an increase of YAP. This is in opposite direction to what we observe in lungs ventilated at low frequency, suggesting that there is no reduction in oxygen levels.

Having said that, we agree with the reviewer that in lungs YAP levels may be affected not only by mechanical signals but also by oxygen levels, and this interplay is certainly not studied here. A full in vivo study of this certainly warrants great interest, but we feel it lays out of the scope of this manuscript. To clarify the scope and limitations of our study, we have now edited our discussion of this issue (text starting in line 283) accordingly.

5. The response provided to address whether or not the cell in the previous Figure 4 are primarily epithelial and endothelial cells is not sufficient. I appreciate the response and find it interesting that the biphasic mechanosensing response is conserved across different cell types in vitro, albeit at different frequency thresholds. However, the need for co-immunostaining remains essential, as it is difficult to interpret the in vivo responses without resolving the cell types that undergo mechanosensing. I recommend referencing doi: 10.1172/JCI125014 for co-immunostaining experiments.

As mentioned in our previous response, we encountered problems in our in vivo

immunostainings that precluded a proper quantification. To resolve this, we resorted to a systematic characterization in vitro of the response of the different cell types, which was in fact a much more time consuming set of experiments. Those experiments show that all relevant cell types (fibroblasts, epithelial cells, endothelial cells) respond to stretch with the same trends, although at different frequency thresholds. We agree that it would be interesting to assess cell-type differences in vivo, but we feel this belongs to a follow-up study centered on in vivo lung responses, which should also address the role of other factors (such as oxygen levels as discussed above). For this manuscript, rat experiments were merely intended to show that a similar response can be observed in vivo. Further, we thank the reviewer for pointing out a relevant reference for immunostaining in lungs, but we note that YAP nuclear to cytoplasmic ratios (which require very clean, high resolution images) were not quantified in that paper.

Reviewer #3 (Remarks to the Author):

The authors have carefully addressed the reviewers' comments and incorporated new aspects into the manuscript. Particularly the addition of the new clutch model is highly valuable and explains the data nicely.

I also appreciate that the authors have attempted to briefly clarify what they mean when they refer to fluidization of cells. However, I still find this term confusing (as did two other reviewers), and there is no experimental data supporting a fluidization of cells. The authors either need to provide experimental proof of a change in the cells' fluidity (which I don't think is necessary as it is not required to support the main findings of this study) or revise their wording more extensively. Forces above a threshold are likely to disrupt the cytoskeleton (including actin), which is supported by the new data presented by the authors, and this disruption may or may not lead to a cell's fluidization. But fluidization is not shown. Hence, I would strongly suggest talking about cytoskeletal disruption rather than about fluidization. The casual use of the word in earlier papers doesn't justify its usage here, and in the initial papers, as the authors correctly pointed out, viscosity had been measured to show a change in fluidity. Particularly, I would suggest to replace the word 'fluidization' by 'cytoskeletal disruption' or something similar in the title and figure captions, to change the abstract accordingly (for example to: 'However, above a given threshold the actin cytoskeleton is disrupted and softens, decreasing loading rates and preventing reinforcement.'). and to introduce the idea of fluidization only in the discussion, as it is one possible interpretation of the data.

We thank the reviewer for the positive assessment of the manuscript and of our revisions. As suggested by this and other reviewers, we now have removed the term "fluidization" except in the discussion. Instead, we refer to the phenomenon as cytoskeletal softening, and also discuss how this is related to cytoskeletal disruption.

Minor points:

I would suggest adding the DAPI images to Fig. 1c.

We have added the images as requested.

Add enlarged images of the blue squares (fits) in Fig. 4b & e.

We have added additional panels with enlarged images as requested.

Reviewer #4 (Remarks to the Author):

The authors have addressed all my comments and have done a good job at addressing the overall key concerns that have emerged throughout the revision process. I keep supporting publication of this work. This is a nice demonstration of a multidisciplinary effort combining classic read outs with YAP nuclear/cytoplasmic localization as proximal and functionally valid read-out in mechanosignaling studies. In this view, it may be appropriate and useful for the lay reader to add at lines 57 or 68 a review that cover this concept more broadly than the sole Dupont et al., 2011 (e.g., Brusatin et al., Nat Mater 2018)

We thank the reviewer for the positive assessment of the manuscript and of our revisions, and for the relevant reference. We have added it as suggested.